# To Freeze or Not to Freeze? Epidemic Prevention and Control in the DSGE Model Using an Agent-Based Epidemic Component

**DOI:** 10.3390/e22121345

**Published:** 2020-11-27

**Authors:** Jagoda Kaszowska-Mojsa, Przemysław Włodarczyk

**Affiliations:** 1Institute of Economics Polish Academy of Sciences, Nowy Świat St. 72, 00-330 Warsaw, Poland; 2Department of Macroeconomics, Faculty of Economics and Sociology, University of Lodz, Gabriela Narutowicza 68, 90-136 Lodz, Poland; przemyslaw.wlodarczyk@uni.lodz.pl

**Keywords:** COVID-19, agent-based modelling, dynamic stochastic general equilibrium models, scenario analyses

## Abstract

The ongoing COVID-19 pandemic has raised numerous questions concerning the shape and range of state interventions the goals of which are to reduce the number of infections and deaths. The lockdowns, which have become the most popular response worldwide, are assessed as being an outdated and economically inefficient way to fight the disease. However, in the absence of efficient cures and vaccines, there is a lack of viable alternatives. In this paper we assess the economic consequences of the epidemic prevention and control schemes that were introduced in order to respond to the COVID-19 pandemic. The analyses report the results of epidemic simulations that were obtained using the agent-based modelling methods under the different response schemes and their use in order to provide conditional forecasts of the standard economic variables. The forecasts were obtained using the dynamic stochastic general equilibrium model (DSGE) with the labour market component.

## 1. Introduction

The early months of 2020 brought the world to an almost complete halt due to the occurrence and outbreak of the SARS-CoV-2 coronavirus, which is responsible for catching the highly lethal COVID-19 disease. Despite the hopes that vigorously developing medical sciences would quickly find an effective remedy, recent months have made it quite clear that such a turn of events is not very likely. As of today, we still lack the proper medical treatments that would significantly increase the survival rate of COVID-19 patients, while a vaccine is still in the testing phase and thus is a rather a remote perspective. In such a situation, the question concerning the shape and range of state interventions whose goal is to reduce the number of infections and deaths has become of paramount importance.

Lockdowns of various scales and compositions were introduced in the majority of the developed economies in order to decrease the transmission of the virus and reduce the hospital occupancy rates. Some countries decided to close their economy abruptly, while others did it on a step-by-step basis. The efficiency and economic impact of the lockdowns have differed depending on the social, cultural and economic characteristics of a given state, which caused their public reception to differ. To date, there are no clear guidelines on how a lockdown policy should be implemented. Therefore, the two major questions that are addressed in the presented paper are:Should we freeze an economy in order to decrease the pace of SARS-CoV-2 transmission?What should the scale and composition of an efficient lockdown policy look like?

Our attempt to explain the macroeconomic consequences of the COVID-19 pandemic and its potential countermeasures is not an exclusive one, as the topic became one of the main topics in the literature of economics. Therefore, we begin our article with a literature review on the impact of the COVID-19 pandemic on public health and the economy. In particular, we focus on the application of the two methodologies that were used as the basis for this article: agent-based models (ABM) and dynamic stochastic general equilibrium models (DSGE), some of which included the Susceptible-Infected-Recovered (SIR) component. In the Section 3, we present our agent-based model, which we used to analyse the scenarios. In Section 4, we present four scenarios of the spread of coronavirus and the regulator’s responses to the epidemiological and economic crises. The ABM model was also used to generate the productivity shocks that fed the DSGE model in the following section. In Section 5, we present the details of the DSGE model, which enabled us to test the macroeconomic consequences of pandemics. In Section 6, the COVID-19 prevention and control schemes are compared in terms of their effectiveness. In Section 7, we discuss the policy implications. Section 8 presents the conclusions.

## 2. Literature Review

The impact of the coronavirus pandemic on a society and the economy has been increasingly explored using very different methodologies recently, among which the predominant ones are the SIR and agent-based approaches. In some cases, the SIR component has been an integral part of more complex computational simulation models.

The SIR model was first successfully implemented into the DSGE model in order to study the COVID-19 pandemic by Eichenbaum et al. [1]. This model gained particular importance and popularity among central bankers in the first phase of the COVID-19 pandemic. The model implied that while a containment policy increases the severity of the recession, it would save roughly half a million lives in the U.S. The article demonstrated that the competitive equilibrium is not socially optimal because infected people do not fully internalise the effect of their economic decisions on the spread of the virus.

With reference to this article, Mihailov [2] estimated the Galí-Smets-Wouters (2012) model with an indivisible labour force for the five major economies that were most affected by the COVID-19 pandemic: the US, France, Germany, Italy and Spain. The author carried out a number of simulations that suggested the recoverable loss of per capita consumption and output in one to two years in an optimistic scenario and the permanent output loss after the shock to the labour supply that will still persist for 10–15 years in a pessimistic scenario.

The equilibrium model with multiple sectors experiencing Keynesian supply shocks, incomplete markets and liquidity that constrained consumers was presented by Guerrieri et al. [3]. The authors opted for closing down the contact-intensive sectors and providing full insurance payments to affected workers as an optimal policy that would enable us to achieve the first-best allocation, despite the lower per-dollar potency of the fiscal policy.

The DSGE methodology, albeit without the explicit SIR component, was also used to examine the impact of the coranavirus outbreak on tourism and to test the policy of providing tourism consumption vouchers for residents [4].

In turn, Bayraktar et al. [5] developed a macroeconomic SIR model of the COVID-19 pandemic that explicitly considered herd immunity, behaviour-dependent transmission rates, remote workers and the indirect externalities of a lockdown. Likewise, using the SIR model, Brotherhood et al. [6] analysed the importance of testing and age-specific policies in the face of the spread of the COVID-19 pandemic. The heterogeneous policy responses in terms of testing, confinements and selective mixing by age group were examined by the authors. Additionally, Toda [7] estimated the SIR component in the context of asset-pricing models paying attention not only to the consequences of the pandemic for the real economy, but also for the financial system.

Parallel with the development of the SIR model and the macroeconomic models with the SIR component, agent-based simulations have also been created. This approach permitted more flexibility in the modelling process. Agent-based models have been used successfully in epidemic modelling in the past [8,9,10]. However, in this paper, we focus only on the models of the spread of the pandemic and its medical and economic consequences that have arisen in the last ten months as they relate directly to the COVID-19 pandemic.

Cuevas [11] elaborated an agent-based model to evaluate the COVID-19 transmission risks in facilities. Under the assumption that each agent has different mobility requirements and contagion susceptibility, Cuevas [11] tested the coexistence of the conditions that need to be imposed and habits that should be avoided in order to reduce the transmission risks.

An interesting combination of the advantages of the ABM and SIR approaches was presented in the model that was developed by Silva et al. [12]. The goal of the COVID-ABS model, which is a new SEIR (Susceptible-Exposed-Infected-Recovered) agent-based model, is to simulate the pandemic dynamics using a society of agents that emulate people, businesses and the government. The authors developed scenarios for social distancing interventions, including scenarios for a lockdown or partial isolation, the use of face masks and the use of face masks together with 50% adherence to social isolation.

The course of the COVID-19 pandemic in regions smaller than countries was studied by Shamil et al. [13]. Their agent-based model was validated by comparing the simulation to the actual data from American cities. The authors’ experiments suggest that contact tracing *via* smartphones combined with a city-wide lockdown is an effective counteractive measure (the reproduction number fell below one within three weeks of the intervention in the scenario that was presented in the paper).

Hoertel et al. [14] examined the effectiveness of a lockdown and the potential impact of post-lockdown measures, including physical distancing, mask-wearing and shielding individuals who are the most vulnerable to a severe COVID-19 infection, on the cumulative disease incidence and mortality and on the occupancy intensive care unit beds. The authors examined the conditions that would be necessary to prevent a subsequent lockdown in France.

Wallentin et al. [15] discussed COVID-19 intervention scenarios for the long-term management of the disease. During the first outbreak of the coronavirus disease, it was restrained in many countries around the world by means of a severe lockdown. Nonetheless, in the second phase of managing the disease, the spread of the virus needs to be contained within the limits that national health systems can cope with. In this study, four scenarios were simulated for the so-called *new normality* using an agent-based model. The authors suggest contact tracing as well as adaptive response strategies that could keep COVID-19 within manageable limits.

Currie et al. [16] addressed the challenges of dealing with the coronavirus pandemic and discussed how simulation modelling could help to support decision-makers in making the most informed decisions. Likewise, Bertozzi et al. [17] discussed the challenges of modelling and forecasting the spread of COVID-19. The authors presented the details of three regional-scale models for forecasting the course of the pandemic. Because they are able to measure and forecast the impact of social distancing, these models highlight the dangers of relaxing the nonpharmaceutical public health interventions in the absence of a vaccine.

Kloh et al. [18] studied the spread of epidemics in low-income settings, given the special socioeconomic conditions surrounding Brazil. The authors applied the agent-based model to simulate how the public interventions could influence the spread of the virus in a heterogeneous population.

The purpose of Maziarz and Zach’s work [19] was to assess the epidemiological agent-based models of the COVID-19 pandemic methodologically. The authors applied the model of the COVID-19 pandemic in Australia (AceMod) as a case study of the modelling practice. The main conclusion was that although epidemiological ABMs do involve simplifications of various sorts, the key characteristics of social interactions and the spread of virus are represented accurately. Kano et al. [20] addressed the relationship between the spread of the virus and economic activities. The agent-based model was used in which various economic activities were considered. The computational simulation recapitulated the trade-off between health and the economic damage that is associated with lockdown measures.

Brottier [21] presented the shortcomings of the Susceptible-Exposed-Infectious-Recovered (SEIR) approach for studying the spread of the virus and emphasised the advantages of the epidemic agent-based models. A more popular-science contribution, which compared the advantages and disadvantages of the SIR and ABM models, was presented by Adam [22]. The strong points of the agent-based approach in epidemic modelling were also highlighted by Wolfram [23]. Because many simple models of disease spread assume homogeneous populations (or population groups) with scalar interaction rates, Wolfram proposed a different approach. The variability between agents in the rate of interactions and the structure of the in-person contact network was included in an agent-based model. The investigation of the properties of this model revealed that there is a critical point in the number of interactions that determines whether everybody gets sick or nobody does. The structure of the contact network and the heterogeneity of agents also matters. The main finding of his article was that reducing the number of interactions between group of agents increases the uncertainty of the outcome, but flattens the curve and reduces the average total number of people who are infected. It is also better to support policies that permit a number of small meetings rather than a few large ones.

Although in our article we attempt to estimate the impact of the pandemic on a society and economy in the short term (up to two years), it is also worth noting that in the literature, the first attempts to estimate long-term effects of the COVID-19 pandemic have been made [24].

## 3. COVID-19 Dynamics—The ABM Approach

We constructed an agent-based model to simulate the spread of the COVID-19 virus and analysed the impact of the pandemic on a society’s overall labour productivity. We then used this model to run four simulations (see Section 3) and estimated the economic impact using the dynamic stochastic general equilibrium model (see Section 4).

In the most basic version, the functioning of the model was defined in six modules, that is, parts of the code. In the first module, the basic parameters and initial conditions were used. The variables and parameters are explained in Table 1, Table 2 and Table 3. The values of these parameters and the probabilities were estimated based on the empirical data and were specific for a given pandemic scenario in a given country. The calibration for a given scenario is explained in Table 4.

The second module created the matrices of a society using the initial parameters. Specifically, the following were created:an M×T matrix H, which recorded the health status of each agent in a society after each iterationan M×T matrix W, which recorded the productivity of each individual in a society after each iterationan M×T matrix A, which recorded the age of each individual in a society after each iterationan M×2T matrix X, which recorded the location of each individual on the map after each iteration (x- & y-coordinates)an M×4 matrix F, which recorded the full data set

We randomly assigned a location, health status and age to each agent (the number of infected people had already been set in the initial conditions).

The third module described the movements of the population (agents) in a closed economy. We used the logic from the cellular automata models (for general information [25,26]). By default, in the basic model, a healthy individual moved in the Moore neighbourhood of a cell (i.e., it is defined on a two-dimensional square lattice and is composed of a central cell and the eight cells that surround it), although this assumption could easily be modified. An infected person (symptomatically and asymptomatically) could move around and continue to infect other agents. When an agent was qualified as deceased, treated or in quarantine, it stopped moving. It is worth noting that when calibrating the scenarios, we used the size of the grid and the number of entities, which provided the actual empirical population density of the selected country.

The fourth module defined the spread of coronavirus in the society. The code analysed the neighbourhood of each agent.

### 3.1. Cases for Healthy Individuals

If there was infected ($s_{t}^{Ind}=2$) or treated person ($s_{t}^{Ind}=3$) in the neighbourhood of a given individual, the healthy person ($s_{t}^{Ind}=1$) could have become infected ($s_{t}^{Ind}=2$) or directly treated in hospital (or put in isolation) ($s_{t}^{Ind}=3$) with a certain probability. If an agent was infected, it did not mean that it had been diagnosed as such. The code first determined whether an agent had become infected (first probability test) and if the test was successful it determined whether this individual had been diagnosed and directed for treatment (second probability test). For agents that had not been infected the program determined whether they had been directed for preventive quarantine ($s_{t}^{Ind}=4$). With a certain probability, a healthy individual could die within one week ($s_{t}^{Ind}=5$). The state transition probabilities in the agent-based epidemic component are described in Figure 1.

### 3.2. Cases for Infected Individuals

For people that were already infected ($s_{t}^{Ind}=2$), the system determined whether they had been directed for treatment ($s_{t}^{Ind}=3$), had died ($s_{t}^{Ind}=5$) or had managed to conquer the virus ($s_{t}^{Ind}=1$). As in the previous case, all of the tests were probabilistic in nature.

### 3.3. Cases for Treated or Infected Individuals in Isolation

Agents that were undergoing treatment ($s_{t}^{Ind}=3$) were reasonably likely to recover ($s_{t}^{Ind}=1$), remain in hospital or in isolation ($s_{t}^{Ind}=3$) or die of the infection ($s_{t}^{Ind}=5$) (with certain probabilities).

### 3.4. Cases for Healthy Individuals in Preventive Quarantine

For the individuals in preventive quarantine ($s_{t}^{Ind}=4$), the system determined the time that an agent had remained in quarantine. After two weeks (two iterations), the agent could be released based on a probabilistic test. The individual could have been healthy after the quarantine ($s_{t}^{Ind}=1$). In addition, a probabilistic test was carried out to determine whether the quarantined person had contracted the virus, for example, as a result of contacts with their immediate family during or at the end of the quarantine (respectively $s_{t}^{Ind}=3$ and $s_{t}^{Ind}=2$). With a very small probability, the individual could also have died during the quarantine ($s_{t}^{Ind}=5$).

It is also worth noting that the probability tests considered the age of an agent. Elderly people have a higher probability of being infected or dying due to coronavirus infection. Changing the health status caused an agent’s productivity to be updated accordingly. A decrease in individuals’ productivity was extensively discussed among the authors, who also consulted with medical specialists. The input data was also consistent with the estimation results from the literature.

In the fifth module, the aggregated values were calculated for each iteration, that is,

the productivity of the societythe number of infected citizens by agethe number of healthy individuals by agethe number of agents undergoing treatment by agethe number of individuals in preventive quarantine by agethe number of deceased by age.

We used this data to determine the productivity shock that fed the dynamic stochastic general equilibrium model.

The last module of the code visualised the results for a given simulation and described the most important information in the output tables for further analysis using the DSGE model (especially the data on the course of the pandemic and the productivity shocks).

## 4. Potential Epidemic Scenarios

As part of the study, we conducted a number of simulations. We present the four most important scenarios that permitted the validity and effectiveness of the restrictions that were introduced in countries in the face of the development of the COVID-19 pandemic to be assessed. In the next part of the article, we also present the impact of the pandemic on the economy using the DSGE model for the following four scenarios.

### 4.1. Scenario 1: The Persistent Spread of the Pandemic under Mild Restrictions

In the first scenario, we analysed the spread of the coronavirus in a country under mild restrictions, that is, we assumed that people with symptoms of the disease were ordered to comply with compulsory home isolation or, in more severe cases, were hospitalised. In both cases, the agents spent at least three weeks there. People who had had contact with an infected person were quarantined with a given probability. The quarantine period was a minimum of two weeks. At the same time, governments decided not to adopt additional restrictions.

In order to simulate this scenario, we assumed that the model worked as was presented in the previous Section 3. In each scenario, one iteration corresponded to a week. The scenarios were carried out for a period of two years (T = 104). In all scenarios, the Monte Carlo methods were used. In order to speed up the simulation, we used 10,000 agents in the model ($N^{Ind}$) and the codes that are available in the external *Comses.net* repository. The results were, however, robust for changing the number of agents all the way up to 1,000,000 and changing the dimensions of the initial grid accordingly ($S_{t}\times S_{t}$ for t=0). We assumed that the initial number of infected individuals was 150. The dimensions of the initial grid were adopted in a such way as to replicate the population density of the country under study. Each individual was characterised by the age. The model also replicated the division of a society in terms of the pre-productive (${\left(Ag\right)}_{t}^{1}$), productive (${\left(Ag\right)}_{t}^{2}$) and post-productive (${\left(Ag\right)}_{t}^{3}$) ages according to official CSO’s statistics. In this scenario, we assumed that the average productivity of an individual who was infected was 0.9, while the average productivity of an agent undergoing treatment in hospital or during home isolation was 0.3. At the same time, the average productivity of a healthy person in preventive quarantine was 0.8. The adopted values were consistent with the results of estimates that were found in the literature. The estimates of the transition probabilities between countries were computed based on the data that was provided by the European Centre for Disease Prevention and Control, the Lancet Commission on COVID-19 and national authorities, see Figure 1 and Table 2, Table 3 and Table 4.

Figure 2 presents the spatial-temporal distribution of healthy ($S_{t}^{Ind}=1$, (*h*)), infected ($S_{t}^{Ind}=2$, (*i*)), treated ($S_{t}^{Ind}=3$ (*l*)), quarantined ($S_{t}^{Ind}=4$ (*k*)) and deceased ($S_{t}^{Ind}=5$ (*d*)) agents at t=1, t=8, t=20 and t=52 respectively. Each point on the coordinate grid represents the agent’s location and its health status. The changes in health statuses are dynamic (in each iteration, that is, in each week, a given agent may have a different health status which depends on agent’s contacts with other people in the neighbourhood). Agents also change their locations.

Figure 3 presents the changes in the labour productivity over time during the pandemic that were under mild restrictions. Each point on the coordinate grid represents the agent’s location and its labour productivity (lower values are marked in darker blue). The disaggregated data was then used to calculate the productivity for a society (for all *t*). When interpreting the Figure 3 and Figure 4, it is worth remembering that people in the pre-productive age and those who were retired had by definition zero productivity. The changes in productivity are dynamic (in each iteration, that is, in each week, a given agent may have a different productivity). Agents also change their locations. A drop in productivity for a person of working age was possible when the person was infected, undergoing treatment or in quarantine (after having direct contact with infected agent in the neighbourhood).

Figure 5 presents a 3D histogram that shows the changes in the number of agents with different health conditions over time. In this scenario, we observed a gradual decrease in the percentage of healthy people. On the other hand, the percentages of people undergoing treatment, quarantined and deceased increased over time. At t=8, 2.79% of the population was infected, 2.12% of the population was hospitalised or in home isolation, 5.13% of the population was healthy, but remained in preventive quarantine, while the mortality rate was marginal and less worrisome (0.02%). After five months, the percentage of healthy people dropped from approximately 98.51% at t=1 to 78.81%, while the percentage of infected agents increased to 6.19%. The percentage of people in preventive quarantine increased to 9.91%. The percentage of hospitalised agents or those who remained in home isolation increased to 5% of population. After one year, the percentage of healthy individuals dropped to 73.34%. The percentage of infected agents remained high at 7.35% of the population at t=52. The percentage of agents in preventive quarantine stabilised at a level of approximately 11.81%, while the percentage of those being treated was 7.22% of population. The percentage of deceased individuals reached 0.28% of the population. After a year, the values stabilised, while the pandemic continued and the negative effects on the economy were visible and (at least partially) permanent.

Figure 15 shows the changes in labour productivity that resulted from the spread of the virus and the adoption of mild restrictions in the form of a quarantine. In the first scenario, the productivity stabilised at approximately 94% of the original value. Thus, there was a permanent decrease in productivity.

### 4.2. Scenario 2: The Spread of Pandemic under Mobility Restrictions

In the second scenario, we analysed the impact of the lockdown on the spread of the virus and on the economy. In this scenario, it was assumed that a very extreme lockdown was introduced for a relatively long period of time (at least two months).

A lockdown was introduced into the model as a mobility restriction that modified the grid and the interactions in the neighbourhood. The grid was dynamically optimised throughout the simulation run. In contrast to the first scenario, in this scenario, the productivity of a healthy agent was not constant and equalled 1. During a lockdown and an open-up phase, the productivity of such an agent was correspondingly lower. The productivity differential reflected the varying degrees of the impact of the pandemic on specific sectors of the economy.

The introduction of an extreme lockdown reduced the long-term decrease in productivity in the economy, see Figure 15. It was also the only solution that returned to the pre-crisis level of productivity within two years without the permanent loss of productivity due to an increase in the number of deaths and the permanent destruction of jobs (which could also lead to an increase in the unemployment rate due to hysteresis).

As it was the case in first scenario, Figure 6 presents the spatial-temporal spread of coronavirus in the society, while Figure 7 illustrates the data on the changes of the labour productivity of the agents over time during the pandemic.

Figure 8 presents a 3D histogram that shows the changes in the number of agents with a different health status over time. At t=8, the percentage of healthy agents in a society was 89.27%, while the percentage of infected agents was 2.88%. At the same time, 2.58% of the population was hospitalised or remained in home isolation and 5.29% was in preventive quarantine. At t=20, there was an increase in the number of infected individuals (up to 6.24%) and those who were placed in preventive quarantine (up to 10.24%). Moreover, 5.96% of the individuals were hospitalised or remained in home isolation and the percentage of the deceased increased to 0.06%. Consequently, only 77.27% of the population was in good health. At t=30, 88.06% of the population was healthy, while 3.70% was infected. Additionally, 4.33% was under preventive quarantine and 3.79% were undergoing treatment. Approximately 0.12% of the population could die. At t=65, the economy and public health returned to normal, i.e., 98.47% of the agents remained healthy, while only 0.36% was infected. A small percentage of the subjects were treated (0.33%), quarantined (0.59%) or died (0.25%).

### 4.3. Scenario 3: The Spread of Epidemic under Gradual Preventive Restrictions

In the third scenario, we analysed the impact of introducing gradually preventive restrictions on a society and the functioning of the economy on the spread of the virus and on the economy. Different types of restrictions were included in the scenario. Specifically, however, various types of mobility restrictions, restrictions that could affect the probability of infection and lockdown were distinguished.

In the second scenario, the grid and the interactions were dynamically adjusted in the neighbourhood (as in the previous scenario), but we also assumed that the restrictions might affect the transition probabilities in the model. The labour productivity of healthy workers during a lockdown and the open-up phase were also optimised as in the previous case. For details, see the code that is available in the external repository *Comses.net*.

About two months after the spread of the virus was identified in a country, preventive measures in the form of the mandatory wearing of masks indoors and a campaign to promote greater hygiene were introduced, see Figure 15. As a result of the information campaign that was conducted, the curve showing the number of new cases flattened out temporarily. At the same time, fewer people required hospitalisation, fewer people were quarantined and the death rate was also much lower. However, due to the behavioural factor, the period of public compliance with the new restrictions did not last longer than a month. From week 11, the agents gradually viewed compliance with the restrictions that had been imposed by the regulator more and more negatively, which increased the number of infections and agents that were placed in preventive quarantine. The increase in the rate of the spread of the virus led to a decrease in the productivity of the individual agents and the entire society.

In response to the increase in the number of cases in society, the regulator introduced new restrictions after approximately a month. In response to the distinction between the restrictions that were imposed on specific areas depending on the incidence rate among the inhabitants of a given area, the incidence curve and, consequently, the productivity curve temporarily flattened. The effectiveness of mobility restrictions within specific areas was relatively low. It was mainly associated with the relatively high level of communication in the zones, the high level of mobility of a society and the need to provide products within the supply chain. As a result, over time, more and more people were infected and more and more zones were issued new restrictions, which turned out to be relatively ineffective.

Due to the alarming number of infections and the general decrease in a society’s productivity, the regulator’s efforts to improve the effectiveness of the countermeasures and regulations were seen. Specifically, mobility restrictions were strengthened, including:local lockdowns, that is, for specific areas of a countrymoderate mobility restrictions in public transportlimiting the number of people participating in assemblies and meetingsan emphasis on remote work in selected sectors of the economy, where this remote work did not reduce the overall productivity of the sectorshybrid preventive measures in the education sector

Once again, the behavioural factor, that is, the degree to which the public adapted to the new operating conditions, was considered. People were less willing to comply with the rules and control schemes that were in place over time. From the 26th week onwards, this caused a renewed increase in the number of infections (also in the number of people in quarantine, undergoing treatment and deaths, respectively) and a decrease in the productivity of a society.

Analysing the data, it was possible to observe a positive temporary impact on the stabilisation of the situation of the measures that had been introduced so far. Therefore, an intensified information campaign was conducted, which was accompanied by tougher penalties for no compliance, which brought about positive results (at least until disinformation campaigns concerning the pandemic on social media and mass media increased).

Along with the growing popularity of disinformation campaigns, the resistance of individuals to comply with the restrictions increase, which was also reflected in protests (protests of companies operating in particularly vulnerable sectors and the anti-COVID-19 movements).

The prolonged epidemiological crisis and the increase in morbidity made the situation of the health care system worse. Problems with the availability of beds and medical equipment in hospitals and the excessive burden on doctors and medical staff grew incessantly. In response to the exponential increase in the number of infections (the number of infections per 1000 inhabitants exceeded the tipping point) and the collapse of the healthcare system, the regulator introduced a total lockdown in the country.

Lockdown caused a decrease in the productivity of all individuals of working age including healthy people. The level of the decrease in productivity depended on the sector in which the agent was employed. Nevertheless, it resulted in a significant reduction in the number of infections and deaths per day. The recovery from a lockdown took place over a longer period of time and was done at different rates by different sectors of the economy, hence, the increase in productivity in the economy was not sudden and was spread over time.

Figure 9 illustrates the changes in the health statuses that resulted from the introduction of preventive restrictions by the social regulator and the appropriate behavioural responses of the agents to the restrictions over time. Figure 10 presents the data on labour productivity of the agents over time during the pandemic in the third scenario.

Figure 11 presents a 3D histogram that shows the changes in the number of agents with different health conditions over time. In this third scenario, we observed successive changes in the percentage of healthy people over the two-year horizon. At t=8, 89.63% of the population was healthy, 2.65% of the population was infected, 1.79% of the population was hospitalised or in home isolation, 5.9% of the population was healthy, but remained in preventive quarantine, while the percentage of deaths in the population reached 0.03%. At t=25, the percentage of healthy individuals decreased to 80.79%. The percentages of infected agents as well as the percentage hospitalised or those in isolation or in preventive quarantine increased (for infected to 5.98%, for hospitalised to 2.87% and for those in preventive quarantine to 10.18%). The percentage of deceased agents reached 0.18% of population. During the lockdown, at t=41, the percentage of healthy individual dropped to 71.01%. At the same time, 7.65% of the agents were infected and 9.14% were undergoing treatment or were in home isolation. Moreover, 11.85% of the population was in preventive quarantine. However, imposing a lockdown had positive medium-term effects on public health and the economy. At t=100, 98.36% of the population was healthy, while only 0.35% were infected and 0.34% were undergoing treatment. The percentage of deceased agents did not exceed 0.5% of the population.

### 4.4. Scenario 4: The Persistent Spread of Pandemic without Restrictions

In the last scenario, we analysed the situation in which the coronavirus spread in the society in a much more aggressive manner and in which the death rate was also higher. In this scenario, we assumed that the regulator had not imposed any restrictions on a society. Specifically, it performed no large-scale testing and did not introduce mandatory isolation for diagnosed individuals or agents who had come into contact with an infected person (preventive quarantine or home isolation). This situation occurred in highly mobile societies with poor-quality healthcare or restricted access to healthcare systems.

In this scenario, we modified the basic model in two ways. On the one hand, we assumed that the virus was more contagious and might have been associated with a higher mortality than was assumed, for example, in the absence of an effective health care system or due to a mutation of the virus. On the other hand, all forms of preventive restrictions and control schemes were excluded from the model. Specifically, in this scenario, agents who had been in contact with an infected person were not required to be quarantined.

In Figure 12, we present the dangerous spread of the virus in a society, while in Figure 13, the changes of the labour productivity of the agents over time are presented. In Figure 14, we present a 3D histogram of the health statuses for the fourth scenario. In this explosive scenario, at t=20, only 62.22% of the population was healthy and almost a quarter of the population was infected (24.54%). There was no preventive quarantine and therefore 11.07% of the population was in hospital or remained at home in less severe cases. The percentage of deceased exceeded 2% of population. The situation got gradually worse. After one year, only 59.46% of the population were healthy, 23.07% of the agents were infected and 10.36% were hospitalised or staying at home. The mortality rate increased significantly. At t=52, 7.11% of population had died due to an infection or comorbidities. If the regulator’s remedial measures were not implemented and the situation had continued to worsen in the following year, we would have seen alarming data on infected and mortality rates and a significant decrease in labour productivity. At t=80, the percentage of infected agents stabilised at 22–23% (it reached 22.51%). However, mainly due to an inefficient health care system, the percentage of hospitalised individuals (or those in home isolation) did not change (10.06%). The death rate increased to 11.55%. This actually showed the scale of the problem and the need for an active public policy from the beginning of the pandemic.

In Figure 15, we present the permanent decrease in productivity in the economy as a result of the increase in agent mortality and infections. When the tipping point of the pandemic was exceeded, crisis management became extremely difficult. An increasing percentage of the population, including those of working age, was infected. This led to downtime in companies and ineffective staff turnover, which caused the more productive and highly skilled sectors to suffer the most. Initially, the exponential trend slowed down gradually. From t=47, there was a practically linear decrease in productivity, which was the result of the gradual (and very slow) development of herd immunity by a society. However, the further decrease in productivity was long-lasting, as we assumed that entities had acquired only temporary immunity, which has been confirmed by the latest research on the coronavirus.

## 5. Macroeconomic Consequences of Pandemics—The DSGE Approach

In order to assess the macroeconomic consequences of the COVID-19 pandemic under different prevention and control schemes, we constructed a DSGE model, which accounted for the most important business cycle characteristics of modern economies. To keep our considerations relatively simple, we adapted the basic model that was proposed by Gali [27] and extended it by introducing a capital accumulation component defined in such a way that it draws heavily from the work of Christiano et al. [28] as well as a labour market component, which was developed along the lines of Gali [27,29] and Gali et al. [30]. In order to make it possible for the model to account for the impact of the COVID-19 pandemic on the economic system being analysed, we also introduced an additional shock, which affected the labour productivity of the agents. This approach enabled us to model the decreases in the availability of employees that were associated with the progress of the COVID-19 spread and the resulting economic disturbances. Below, we present and discuss the most important characteristics of the macroeconomic model that was used in our further analyses and its calibration.

The model assumed that an economy was populated by a unit mass *continuum* of households that maximised their utility levels by solving the following optimisation problem:(1)$$\max E_{0}\left\{\sum_{t=0}^{\infty }\beta^{t}\left[U\left(C_{t},N_{t}\right)\right]\right\},$$
where: $E_{0}$ is a rational expectations operator that represents the information that a household has in period 0; β is a discount factor so that β∈[0;1]; $C_{t}$ is the value of a household’s total consumption in period *t*; $N_{t}$ is the amount of labour that was provided by a household in period *t*; $U\left(C_{t},N_{t}\right)$ is a twice differentiable, instantaneous utility function and $\frac{\partial U\left(C_{t},N_{t}\right)}{\partial C_{t}}>0$, $\frac{\partial^{2}U\left(C_{t},N_{t}\right)}{\partial^{2}C_{t}}\leq 0$ and $\frac{\partial U\left(C_{t},N_{t}\right)}{\partial N_{t}}>0$, $\frac{\partial^{2}U\left(C_{t},N_{t}\right)}{\partial^{2}N_{t}}\leq 0$ that represents the diminishing marginal utilities of consumption and labour. The utility function is of the King et al. [31] type, namely: $U\left(C_{t},N_{t}\right)=\ln{\tilde{C}}_{t}-\epsilon_{t}^{\chi }\frac{N_{t}^{1+\varphi }}{1+\varphi }$, where $\epsilon_{t}^{\chi }$ is an exogenous preference shifter that represents the impact of a labour supply shock governed by an AR(1) process of the form: $\ln\epsilon_{t}^{\chi }=\rho_{\chi }\ln\epsilon_{t-1}^{\chi }+\xi_{t}^{\chi }$, $\xi_{t}^{\chi }∼i.i.d.N\left(0,\sigma_{\chi }^{2}\right)$, $\rho_{\chi }\in \left[0;1\right]$ and φ>0 is the inverse of the Frisch elasticity of the labour supply. Following the empirical models of Christiano et al. [28], Smets and Wouters [32] and Gali et al. [30] and more fundamentally the seminal paper by Abel [33], it was assumed that households’ consumption is characterised by the habit persistence, which is determined by an external habit formation of the form: ${\tilde{C}}_{t}\equiv C_{t}-hC_{t-1}$, where h∈[0,1] is the habit persistence parameter and $C_{t-1}$ is the value of lagged aggregate consumption.

Households’ income comes from work (its differentiated types are indexed with *i*) and lump-sum transfers. It is used in order to finance current consumption, which involves the purchase of diversified goods that are produced by companies (with types indexed with *z*) or postponing consumption and buying one-period risk-free government bonds (the so-called *Arrow securities*). In order to make our DSGE model closer to the standard economic representations of the production process, we also included capital into our considerations. The physical stock of capital is owned and maintained by the households who rent its services to the companies. The capital market is perfectly competitive and the nominal capital rental rate is given by $R_{t}^{k}$. Following the discussion in Christiano et al. [28] and Christiano et al. [34], the capital accumulation process is represented by equation:(2)$$K_{t+1}=\left[1-\frac{\phi_{k}}{2}{\left(\frac{I_{t}}{I_{t-1}}-1\right)}^{2}\right]I_{t}+\left(1-\delta \right)K_{t},$$
where: $\phi_{k}>0$ is the capital adjustments costs’ scaling parameter and δ∈(0;1) is the capital depreciation rate.

The intertemporal budget constraint of a household, which equates income with spending is written as:(3)$$\int_{0}^{1}C_{t}\left(z\right)P_{t}\left(z\right)dz+I_{t}+Q_{t}B_{t}\leq B_{t-1}+\int_{0}^{1}W_{t}\left(i\right)N_{t}\left(i\right)di+R_{t}^{k}K_{t}+Div_{t}-T_{t},$$
where: $C_{t}\left(z\right)$ and $P_{t}\left(z\right)$ respectively denote consumption and the price of the *z*-th type goods, $C_{t}={\left(\int_{0}^{1}C_{t}{\left(z\right)}^{\frac{\varepsilon_{c}-1}{\varepsilon_{c}}}dz\right)}^{\frac{\varepsilon_{c}}{1-\varepsilon_{c}}}$; $N_{t}\left(i\right)$ and $W_{t}\left(i\right)$ are the *i*-th type labour wage levels in period *t*; $\varepsilon_{c}\geq 1$ describes the elasticity of the substitution between different types of goods; $Q_{t}$ denotes the price of the Arrow securities; $B_{t}$ is the number of risk-free government bonds purchased at a discount by a household in period *t*; $Div_{t}$ is the value of all of the dividends received by households from companies; and $T_{t}$ is the net value of all lump-sum taxes paid and transfers received by a representative household.

Solving the households’ optimisation problem requires tackling the problem of the optimal allocation of expenditures among the different types of goods, which results in: $C_{t}\left(z\right)={\left[\frac{P_{t}\left(z\right)}{P_{t}}\right]}^{-\varepsilon_{c}}C_{t}$, $\int_{0}^{1}P_{t}\left(z\right)C_{t}\left(z\right)dz=P_{t}C_{t}$, $P_{t}={\left(\int_{0}^{1}P_{t}{\left(z\right)}^{1-\varepsilon_{c}}dz\right)}^{\frac{1}{1-\varepsilon_{c}}}$ and in the transversality condition, which is given by: $\lim_{T\to \infty }\beta^{T}E_{t}\left\{\frac{B_{T}}{C_{T}}\right\}\geq 0$.

The model accounts for the existence of wage rigidities. It is assumed that households provide differentiated labour services (indexed by *i*) and that the level of wages is determined by trade unions that specialise in supplying only a given type of labour. Each of the unions is an effective monopolist as the supplier of a given type of labour. Because of their position, they can demand wage rates that exceed the marginal rate of substitution between consumption and leisure by a mark-up that is indicative of their market power. The renegotiation of employment contracts with entrepreneurs is costly and subjected to some restrictions, similar to those that were introduced by the Calvo [35] pricing scheme. Namely, only the exogenously determined, randomly selected group of trade unions given by $1-\theta_{w}$, where $\theta_{w}\in \left[0;1\right]$, can re-optimise wages in a given period by choosing $W_{t}^{*}$. The group is large enough for its decisions to have impact on the aggregate nominal wage rate, which is given by $W_{t}$. When taking decisions about the level of wages, trade unions consider the consumption choices of households that supply a given type of labour and take the maximisation of the households’ utility as their ultimate goal. Assuming that all of the households are identical results in the following symmetrical problem:(4)$$\max_{W_{t}^{*}}E_{t}\left\{\sum_{k=0}^{\infty }{\left(\beta \theta_{w}\right)}^{k}U\left(C_{t+k|t},N_{t+k|t}\right)\right\},$$
(5)$$N_{t+k|t}={\left(\frac{W_{t}^{*}}{W_{t+k}}\right)}^{-\varepsilon_{w}}\int_{0}^{1}N_{t}\left(z\right)dz,$$
(6)$$P_{t+k}C_{t+k|t}+I_{t+k|t}+Q_{t+k}B_{t+k|t}\leq B_{t+k-1|t}+W_{t+k}N_{t+k|t}+R_{t+k}^{k}K_{t+k|t}+Div_{t+k}-T_{t+k},$$
where $C_{t+k|t},W_{t+k|t}^{*},B_{t+k|t},I_{t+k|t},K_{t+k|t}$ denote the level of consumption, nominal wages, risk-free government bonds, investments and capital selected by a household or a trade union that re-optimises wages in period *t* and keeps them unchanged up to and including period t+k, respectively. The FOC of the trade union’s optimisation problem is given by:(7)$$\sum_{k=0}^{\infty }{\left(\beta \theta_{w}\right)}^{k}E_{t}\left\{N_{t+k|t}U\left(C_{t+k|t},N_{t+k|t}\right)\left[\frac{W_{t}^{*}}{P_{t+k}}-\frac{\varepsilon_{w}}{\varepsilon_{w}-1}MRS_{t+k|t}\right]\right\}=0,$$
where $MRS_{t+k|t}=-\frac{U_{N}\left(C_{t+k|t},N_{t+k|t}\right)}{U_{C}\left(C_{t+k|t},N_{t+k|t}\right)}$ is the marginal rate of the substitution of the households/labour unions that selected a nominal wage level in period *t* and kept it unchanged up to and including period t+k. The average wage level in this case is given by: $W_{t}={\left[\theta_{w}{\left(W_{t-1}\right)}^{1-\varepsilon_{w}}+{\left(1-\theta_{w}\right)}^{1-\varepsilon_{w}}\right]}^{\frac{1}{1-\varepsilon_{w}}}$.

As well as choosing the optimal wage level, households also make decisions about labour supply. The decisions are crucial from the perspective of the unemployment component because unemployment is determined by comparing the labour supply and labour demand that arise from the production needs of firms. That part of the model is developed according to the framework proposed by Gali [29]. It assumes that each of the infinitely many households that are indexed by g∈[0;1] have an unlimited number of members given by a *continuum* of size one [36]. Household members provide diversified labour services that involve specific levels of disutility, which is given by $\epsilon_{t}^{\chi }j^{\varphi }$, where $\epsilon_{t}^{\chi }>0$ is an exogenous labour supply shock that affects all of the household members in exactly the same way, φ>0 denotes the elasticity of the marginal disutility from labour between the household members, and *j* stands for disutility from labour, which is normalised so that j∈[0,1]. Therefore, the economy has infinitely many units that are defined in the g×i×j space with the dimensions of [0,1]×[0,1]×[0,1] and that are indexed by vector (g,i,j).

Labour market participation decisions are taken individually by household members with a view to maximising a household’s utility from consumption and leisure. In considering whether or not to work, household members take into account the households’ choices concerning the optimal level of consumption and the trade unions’ decisions about the level of real wages. In other words, they treat the values of all of the variables other than labour supply as given and assume that all job seekers will find employment. Therefore, they need to solve the following optimisation problem:(8)$$\max E_{L_{t}\left(g,i,j\right)}\left\{\sum_{t=0}^{\infty }\beta^{t}\left[U\left(C_{t},\epsilon_{t}^{\chi }j^{\varphi }L_{t}\left(g,i,j\right)\right)\right]\right\},$$
(9)$$P_{t}C_{t}+Q_{t}B_{t}+I_{t}\leq B_{t-1}+W_{t}\left(i\right)L_{t}\left(g,i,j\right)+R_{t}^{k}K_{t}+Div_{t}-T_{t},$$
where $L_{t}\left(g,i,j\right)$ is a dummy variable that has a the value of 0 when an individual chooses not to work and 1 when they enter the labour market.

From the FOC of the optimisation problem that is defined in Equations (Equation 8) and (Equation 9) it follows that individuals will be interested in entering the labour market as long as $\frac{W_{t}\left(i\right)}{P_{t}}\geq \frac{\epsilon_{t}^{\chi }j^{\varphi }}{U_{C,t}}$, which means that the marginal income from work is greater than its marginal disutility, which is expressed by the units of consumption. If disutility from work is ordinal and its increments between individuals doing the same type of work are constant, which means that the increments are evenly distributed over the j∈[0;1] interval, then it is the disutility of the marginal employee doing a given type of work that determines the rate of economic activity and, consequently, the size of labour supply in the analysed model, $L_{t}\left(i\right)$. Because of the previous assumptions about the homogeneity of households and indivisibility of labour, the above problem is symmetrical and its solution for the aggregate level is the same as the one that is obtained by aggregating the results for individual units and households. This allows the aggregate labour supply equation to take the form of:(10)$$\frac{W_{t}}{P_{t}}=\epsilon_{t}^{\chi }{\tilde{C}}_{t}L_{t}^{\varphi },$$
where: $W_{t}\equiv {\left(\int_{0}^{1}W_{t}{\left(i\right)}^{1-\varepsilon_{w}}di\right)}^{\frac{1}{1-\varepsilon_{w}}}$ and $L_{t}\equiv \int_{0}^{1}L_{t}\left(i\right)di$.

In keeping with Gali [27,29] or Gali et al. [30], we assumed that the unemployment rate ($UR_{t}$) was equivalent to the share of unemployed (understood as the excess of labour supply over demand), ($U_{t}\equiv L_{t}-N_{t}$) in the aggregate labour supply. After simple transformations, we have:(11)$$UR_{t}\equiv \frac{L_{t}-N_{t}}{L_{t}}=1-\frac{N_{t}}{L_{t}}.$$
By combining the aggregate labour supply condition from Equation (Equation 10) with the definitions of the marginal rate of substitution and the actual wage mark-up ($M_{w,t}$), we get:(12)$$UR_{t}=1-M_{w,t}^{-\frac{1}{\varphi }}.$$
The framework enabled us to obtain a simple relationship that associates the development of the unemployment rate with changes in the level of wage markup. The larger the actual mark-up over the perfectly competitive wage, the higher the unemployment rate.

The model assumes that the economy being considered has a unit mass *continuum* of firms that produce different categories of goods with both firms and goods being indexed by z∈[0;1]. To produce output $Y_{t}$, firms use identical technology, which is described by the standard Cobb-Douglas production function:(13)$$Y_{t}\left(z\right)=A_{t}K_{t}{\left(z\right)}^{A}{\left[\epsilon_{t}^{N}N_{t}\left(z\right)\right]}^{1-A},$$
where: $A_{t}$ is a technological shock of the form: $\ln A_{t}=\ln\epsilon_{t}^{a}=\rho_{a}\ln\epsilon_{t-1}^{a}+\xi_{t}^{a},\;\xi_{t}^{a}∼i.i.d.N\left(0;\sigma_{a}^{2}\right),\;\rho_{a}\;\in \;\left[0;1\right]$; A∈[0;1]. In order to account for the impact of COVID-19 spread on an economy we endowed the production function of the model with the labour productivity shock that affects uniformly all of the companies. The shock takes the form of: $\ln\epsilon_{t}^{N}=\rho_{N}\ln\epsilon_{t-1}^{N}+\xi_{t}^{N},\;\xi_{t}^{N}∼i.i.d.N\left(0;\sigma_{N}^{2}\right),\;\rho_{N}\;\in \;\left[0;1\right].$ We believe that, this is justified in order to treat COVID-19-caused disturbances as a transitional random shock, because from the point of view of a company, their occurrence results in a sudden and unpredictable change in the economic conditions for which firms can only react with a considerable delay. In the majority of cases it does not make any difference whether these disturbances were incurred by the development of the pandemic itself or as a result of the introduction of state-operated prevention and control schemes, as the dynamics of the pandemic and the speed with which the decisions are taken leaves only a small margin for reaction. On the other hand, due to the relatively low mortality of people in the working age it does not affect the economic conditions in the long run considerably and finally vanishes. The proposed specification which treats the COVID-19-related shock as a labour productivity shock enabled us to envisage the consequences of a change in the availability of employees due to being sick, hospitalised, quarantined or in domestic isolation as well as due to introduction of remote work organisation, which might either prevent them from working at all or significantly reduce their individual efficiency. It should be noted that in each of these cases, employees do not provide a high standard of work, although they are still working for a given company and are being remunerated on a fairly standard basis. As such, the COVID-19 shock should not be considered a labour supply shock, which pushes part of the labour force into inactivity, but rather a labour productivity shock, which makes some of the employees unproductive or not fully productive, while keeping them within a formal employment relationship.

It is further assumed that firms choose prices of goods according to the Calvo [35] formalism. In a given period, they can be re-optimised only by a randomly determined group of firms that are proportional to $1-\theta_{p}$ (where $\theta_{p}\;\in \;\left[0;1\right]$). As a result, $\theta_{p}$ becomes a natural index of price rigidity. Each company re-optimising prices maximises its profit over the predicted period of price validity, which is given by $\frac{1}{1-\theta_{p}}$. Therefore, firms need to solve the following problem:(14)$$\max_{P_{t}^{*}}\sum_{k=0}^{\infty }\theta_{p}^{k}E_{t}\left\{\Lambda_{t,t+k}\left[P_{t}^{*}Y_{t+k|t}-\Psi_{t+k}\left(Y_{t+k|t}\right)\right]\right\}$$
subject to:(15)$$Y_{t+k|t}={\left[\frac{P_{t}^{*}}{P_{t}}\right]}^{-\varepsilon_{c}}Y_{t+k},$$
where: $Y_{t+k|t}\geq C_{t+k|t}+I_{t+k|t}$; $Y_{t+k|t},C_{t+k|t},I_{t+k|t}$, respectively, denote the amount of output supplied, consumption to be met and investments that are introduced by a company re-optimising its prices in period *t* and keeping them unchanged up to and including period t+k; $P_{t}^{*}$ is the price that is chosen by companies that re-optimise prices in period *t*; $\Psi_{t}\left(Y_{t+k|t}\right)$ is the nominal marginal cost of a company that re-optimises prices in period *t* and keeps them unchanged up to and including period t+k; and $\Lambda_{t,t+k}=\beta^{k}E_{t}\left\{\frac{C_{t}P_{t}}{C_{t+k}P_{t+k}}\right\}$. Because all of the companies that re-optimise prices in a given period take the same decision, the optimisation problem is symmetrical and easy to solve. The aggregate price level is then given by: $P_{t}={\left[\theta_{p}P_{t-1}^{1-\varepsilon_{c}}+\left(1-\theta_{p}\right)P_{t}^{*\;1-\varepsilon_{c}}\right]}^{\frac{1}{1-\varepsilon_{c}}}$.

Household members provide firms with diversified labour services, which are indexed by i∈[0;1]. In such a case, a firm’s demand for labour can be expressed using the *Armington’s aggregator* ([37]; which is also known as *Dixit-Stiglitz’s aggregator*) given by:(16)$$N_{t}\left(z\right)={\left(\int_{0}^{1}N_{t}{\left(i,z\right)}^{\frac{\varepsilon_{w}-1}{\varepsilon_{w}}}di\right)}^{\frac{\varepsilon_{w}}{\varepsilon_{w}-1}},\forall \;\;i,z\in \left[0,1\right].$$
The level of employment in firms is assessed using a two-stage budgeting procedure [38,39] with which the optimal allocation of expenditures to different types of labour can be defined for every allowable level of costs, and then a firm’s total demand for labour, which is conditional on the previous solution. Consequently, the following labour demand schedule is obtained:(17)$$N_{t}\left(i,z\right)={\left[\frac{W_{t}\left(i\right)}{W_{t}}\right]}^{-\varepsilon_{w}},\forall \;\;i,z\in \left[0;1\right],$$
where $W_{t}\left(i\right)$ is the real wage amount paid for the *i*-th type of labour and $W_{t}={\left[\int_{0}^{1}W_{t}{\left(i\right)}^{1-\varepsilon_{w}}di\right]}^{\frac{1}{1-\varepsilon_{w}}}$ represents the aggregate wage level in the economy. Based on the functions presented above, we also get the expression: $\int_{0}^{1}W_{t}\left(i\right)N_{t}\left(i,z\right)di=W_{t}N_{t}\left(z\right)$.

The proposed model becomes complete with the introduction of additional market clearing conditions. The clearing of the goods on the market requires that $Y_{t}\left(z\right)=C_{t}\left(z\right)+I_{t}\left(z\right)$. Knowing that $Y_{t}={\left(\int_{0}^{1}Y_{t}{\left(z\right)}^{\frac{\varepsilon_{c}-1}{\varepsilon_{c}}}dz\right)}^{\frac{\varepsilon_{c}}{1-\varepsilon_{c}}}$ and $I_{t}=\int_{0}^{1}I_{t}\left(z\right)dz$ we can easily show that $Y_{t}=C_{t}+I_{t}$. When prices are sticky, the labour market is cleared at a lower level of employment than when they were perfectly elastic. The labour market clearing is described by the following equation:(18)$$N_{t}=\int_{0}^{1}\int_{0}^{1}N_{t}\left(z,i\right)\;di\;dz=\int_{0}^{1}N_{t}\left(z\right)\int_{0}^{1}\frac{N_{t}\left(z,i\right)}{N_{t}\left(z\right)}\;di\;dz.$$
Using the appropriate labour demand functions and the expression for the production function of an individual firm, we obtain:(19)$$\begin{array}{c} N_{t}=\int_{0}^{1}N_{t}\left(z\right)\int_{0}^{1}{\left[\frac{W_{t}\left(i\right)}{W_{t}}\right]}^{-\varepsilon_{w}}\;di\;dz=\Delta_{w,t}\int_{0}^{1}N_{t}\left(z\right)\;dz=\Delta_{w,t}\int_{0}^{1}\epsilon_{t}^{N}{\left(\frac{Y_{t}\left(z\right)}{A_{t}K_{t}{\left(z\right)}^{A}}\right)}^{\frac{1}{1-A}}\;dz\\ =\Delta_{w,t}\int_{0}^{1}\epsilon_{t}^{N}{\left(\frac{{\left[\frac{P_{H,t}\left(z\right)}{P_{H,t}}\right]}^{-\varepsilon_{c}}Y_{t}}{A_{t}K_{t}^{A}}\right)}^{\frac{1}{1-A}}\;dz=\Delta_{w,t}\Delta_{p,t}\epsilon_{t}^{N}{\left(\frac{Y_{t}}{A_{t}K_{t}^{A}}\right)}^{\frac{1}{1-A}}, \end{array}$$
where: $K_{t}^{A}=\int_{0}^{1}K_{t}{\left(z\right)}^{A}\;dz$; $\Delta_{p,t}=\int_{0}^{1}{\left[\frac{P_{H,t}\left(z\right)}{P_{H,t}}\right]}^{-\frac{\varepsilon_{c}}{1-A}}\;dz$ is the measure of the domestic price dispersion and $\Delta_{w,t}=\int_{0}^{1}{\left[\frac{W_{t}\left(i\right)}{W_{t}}\right]}^{-\varepsilon_{w}}\;di$ is the measure of wage dispersion. It follows easily from Equation (Equation 19) that the aggregate production function is given by
(20)$$Y_{t}=\frac{A_{t}K_{t}^{A}{\left(\epsilon_{t}^{N}N_{t}\right)}^{1-A}}{{\left(\Delta_{p,t}\Delta_{w,t}\right)}^{1-A}},$$
whereas the real marginal cost can be specified as
(21)$$RMC_{t}=\frac{\partial RTC_{t}}{\partial Y_{t}}=\frac{W_{t}}{P_{t}}\frac{{\left(\Delta_{p,t}\Delta_{w,t}\right)}^{1-A}{\left(\epsilon_{t}^{N}N_{t}\right)}^{A}}{\left(1-A\right)A_{t}K_{t}^{A}}.$$

In order to close the model, we need one additional equation that explains the specification of the nominal interest rate, which is called a monetary policy rule. It is usually assumed that monetary authorities adopt a policy whose goal is to prevent prices and output from deviating too much from the steady-state values, which can be described using the following Taylor-type rule:(22)$$\frac{R_{t}}{R}=\Pi_{t}^{p\;\phi_{\pi }}{\left(\frac{Y_{t}}{Y}\right)}^{\phi_{y}}e^{\epsilon_{t}^{M}},$$
where $R_{t}$ is the nominal interest rate; $\Pi_{t}^{p}=\frac{P_{t}}{P_{t-1}}$ is the inflation rate; $\phi_{\pi }$ and $\phi_{y}$ are the parameters that describe the monetary authorities’ reaction to any price and output deviations from their steady state values, and $\epsilon_{t}^{M}=\rho_{M}\epsilon_{t-1}^{M}+\xi_{t}^{M},\;\xi_{t}^{M}∼i.i.d.N\left(0;\sigma_{M}^{2}\right),\;\rho_{M}\;\in \;\left[0;1\right]$ is a monetary policy shock.

The full set of the equilibrium conditions of the DSGE model is obtained by combining and transforming the equations that were obtained as solutions to the aforementioned optimisation problems. The model is expressed in weekly terms and is calibrated so that it matches the standard stylised facts concerning the business cycle characteristics of developed economies. As a result, we obtain a model, that successfully reproduces the results of the existing empirical research such as, for example, the estimated model of Christiano et al. [40]. As the model is expressed in weekly terms, which is necessary in order to reproduce the pace and timing of the COVID-19 epidemic, which is quite rare in DSGE research, the actual values that were used in the calibration might cause some reflection. In what follows, we assume the discount factor β=0.9996, which results in the steady-state interest rate of 2.1% in annual terms. Following Christiano et al. [40] and Gali [27] we set the expected duration of prices and wages to 52 weeks, that is, 4 quarters, which makes $\theta_{p}=\theta_{w}=0.9807$. Similar as in Gali [27], we assumed that $\epsilon_{w}=4.52$ and φ=5. As a result the steady-state unemployment rate (which in the case of the analysed model might be under certain restrictions that can be identified with the natural unemployment rate) takes the value of 4.8%. Although the habit persistence parameter, *h*, is set at a relatively high level of 0.9, it seems to be acceptable if we take into account the fact that the model is expressed in weekly terms. We should expect that consumption is characterised by relatively high week-to-week inertia. The capital share in production, which is given by α is given at the level of 0.25. In order to obtain the appropriate reactions of capital and investment to the changes of economic conditions, we assumed that $\phi_{k}=8$, which is relatively close to the assessments that were provided by Christiano et al. [40], and δ=0.05, which is the level that permits the model to be identified. The parameters of the Taylor rule are taken at the level of: $\phi_{\pi }=0.115$ and $\phi_{y}=0.0096$, which enables us to obtain a rule that is consistent with the traditional version of the rule that takes the values of 1.5 and 0.125 in quarterly terms, respectively. Finally, the autoregressive parameters of the shocks are selected in order to obtain the satisfactory duration of shocks in weekly terms. As a result, we assumed: $\rho_{a}=\rho_{\chi }=\rho_{N}=0.99$ and $\rho_{M}=0.965$. The proposed calibration ensures that the model will be identified and also fulfills the Blanchard-Kahn conditions. Due to the fact that this DSGE model is microfounded and have deep parameters that are invariant to changes in economic policy, in principle it is not subject to the Lucas critique. However, for more general discussion about DSGE models and the Lucas critique, see Hurtado [41]. The resistance to the Lucas and Velupillai’s critiques of core and periphery models (in particular DSGE and ABM models) was also studied in Kaszowska et al. [42]. The model was expressed and solved in non-linear terms, that is, we did not log-linearise it around the steady state.

## 6. COVID-19 Prevention and Control Schemes—Efficiency Comparison

In this part of the paper we use the labour productivity paths (Figure 15) that were generated from the agent-based epidemic component of Section 3 in order to obtain conditional forecasts of the standard macroeconomic indicators: output, capital, investments and unemployment rate. The forecasts come from the DSGE model that was described in Section 5. Its calibration uses the standard values that are characteristic for a developed economy. The analyses were based on four scenarios that introduced different prevention and control schemes (as presented in Section 4). All of the results are expressed as the relative difference from the steady state value. The analyses were performed within a two year horizon, which is the minimum that is necessary in order to produce a vaccine or establish an efficient cure for the virus. The presented results constitute the mean of 10,000 simulations of the model. Our discussion concludes with a brief analysis of the robustness of the obtained estimates.

The results of the forecasts that were performed are presented in Figure 16. Their analysis showed that the scenarios can easily be divided into two groups, that produce similar economic trends. The first group consists of Scenarios 1 and 4, which resulted in the occurrence of negative economic trends that persisted in an economy in the medium or even long term. The other group is composed of Scenarios 2 and 3. In that case the economic distortions were relatively short lived, but their impact was greater.

The first group consisted of the scenarios that assumed that the government permitted the persistent spread of the disease by introducing only general sanitary restrictions that were willingly undertaken and obeyed by the society (Scenario 1) or by not introducing any restrictions at all and hoping that the propagation of the virus would finally cease at some point (Scenario 4). Both of these approaches resulted in a relatively high share of people who were either infected or were in quarantine, which translated into a persistent decrease in the productivity of labour, which stabilised at the level of approximately 92% of the full capacity or, in the case of unconstrained spread scenario, exhibited a continuous downward trend that reached the level of 80% within the two years after the beginning of the pandemic. This behaviour of labour productivity translated into the way in which the other variables responded to the shock. In the case of Scenario 1, output decreased by at least 2.5% and towards the end of the sample, it stabilised at 98% of its steady state value. Additionally, there was a permanent decrease in capital and investment of approximately 10%. The unemployment rate increased by 6 pp. in the first year of the epidemic and stabilised at 5 pp. above the steady state later on. This meant that the actual unemployment rate was approximately 9%. In Scenario 4, the changes were much deeper. Although the output initially decreased by approximately 4%, after a short stabilisation, it continued in a downward trend and reached approximately 94% of the initial capacity. Capital and investments also decreased as the persistent decrease of output discouraged enterprises from undertaking development activities. The unemployment rate increased by as much as 15 pp. within the first two years of the pandemic. This resulted in a high social cost, because the actual unemployment rate reached 20%. The costs of Scenario 4, which are presented above only include its short- and medium term consequences and do not include the potential long term loss of human capital that resulted from the high death rate. Including the long term consequences into our assessment would, however, have resulted in the deterioration of the overall balance, which proves that the strategy of no reaction should not be considered to be a viable alternative by the government. Moreover, the solution of Scenario 1, however tempting it was, turned out to be extremely difficult to implement in practice. Only a few countries successfully curtailed the levels of COVID-19 infections solely *via* the use of general sanitary restrictions. In the majority of countries, the society found it extremely difficult to reduce the number of social contacts and to isolate from families and friends.

When assessing the efficiency of the second group of measures that might be introduced in order to limit the transmission of a virus, which consisted of different lockdown schemes, it can easily be observed that, when they were applied with an appropriate strength, they were able to stabilise the number of infections. In our baseline scenario, we assumed that a lockdown consisted of a decrease in professional activity by an average of 15 pp for a period of two months. This rough assessment reflected the experience of the first wave of lockdowns that were introduced in the spring of 2020, when it was determined that a vast majority of the jobs that: are performed in the open air, where the risk of infection is reduced; closed spaces that can be arranged so as to decrease direct contact of workers, such as factories or office buildings; or those that can be performed remotely, did not suffer from a significant decline or delays. The jobs that were badly affected by a lockdown policy were those that rely on direct contact with a customer or the direct contact with a group of people in a closed space, including: shops, restaurants, hotels and the tourist infrastructure as well as cultural and educational institutions. As a result, only a relatively small part of an economy was completely closed during a lockdown. Our assessment of the severity of a lockdown seems to be in line with the actual economic records, as it enabled us to generate a decrease in output of about 8% compared to the OECD average of a 9.8% decrease in the second quarter of 2020. Furthermore, in order to separate the impact of a single lockdown on an economic system, we assumed that after a lockdown, individuals would behave according to standard sanitary restrictions.

Our results clearly show that a lockdown not only results in a decrease of output, but also in a drastic decrease in investments. At the same time, there was only a moderate decrease in the capital level, which resulted from the fact that an economic downturn was strictly limited in time. Finally, the unemployment rate temporarily increased and reached relatively high levels. What is important, is that the depth of a recession that was caused by a lockdown did not depend on the style in which a lockdown was introduced. No matter whether Scenario 2 was followed and a lockdown was introduced immediately, or was done gradually as in Scenario 3, the macroeconomic variables decreased by almost the same amount. What is extremely important is the duration of the economic downturn that was caused by a lockdown. It is clearly visible that a lockdown that lasted for two months resulted in a decrease in economic activity that disappeared after 24 weeks, that is, within half of a year, when the economic recovery began with a period of increased activity.

According to our results, there is a clear trade-off between the duration and severity of a recession that is caused by an epidemic. If we decide to shape our policy according to Scenarios 2 or 3 the changes in economic activity might be abrupt but short-lived. In the case of Scenarios 1 or 4, the decrease in economic activity might not be as deep, but would be rather permanent.

The results of the analyses of Scenarios 2 and 3 also enabled us to compare the efficiency of an immediate or a gradual lockdown. It was determined that the widespread opinion that we should introduce lockdowns gradually so as not to disrupt an economic system was not confirmed by the formal economic modelling. Gradual lockdowns, which are initially too weak to prevent the spread of a disease already curtail economic activity, thus decreasing the level of output below its steady state level. While they do not change the dynamics of an epidemic, they unnecessarily prolong the duration of an intervention and thus are suboptimal compared to an immediate lockdown.

One of the most important assumptions that underlies the results presented in this section concerns the impact of the decrease in labour productivity during the lockdown phase, which was chosen arbitrarily in order to recreate an economic reaction that was observed in actual economic data from the 2nd quarter of 2020. In order to test the robustness of our conclusions, we present the estimates of Scenario 2 for the case in which the lockdown decreased productivity at the baseline level of 85% of its steady state value, together with the results obtained under the assumption that it decreased to the level of 10 pp. and 20 pp. below its baseline value. The labour productivity paths that were simulated under these scenarios are presented in Figure 17. Conditional forecasts of the macroeconomic variables that were obtained for these productivity shocks are presented in Figure 18.

An analysis of the outcomes enabled us to infer that despite the fact that the deeper changes in labour productivity caused more pronounced swings of macroeconomic variables, there was no evidence that such changes affected the duration of a recession that was triggered by a lockdown. This conclusion is of major importance as it confirms our finding concerning the trade-off between the severity and duration of the economic consequences of an epidemic and thus validates it as a foundation for an efficient prevention and control policy.

## 7. Policy Implications

The results of the analyses that were performed in Section 6 enabled us to draw important conclusions with respect to the range and composition of the desired prevention and control schemes whose goal was to minimise the negative economic consequences of an epidemic. They support the use of lockdowns as an efficient tool in the fight against the spread of a disease and indicate the benefits of their immediate introduction. As such, our conclusions are mostly at odds with the widespread conviction that we should strive to keep at least part of an economy open at any cost.

Under these circumstances we should consider a policy that is based on alternating the use of lockdowns and periods of mild restrictions as a viable alternative to the currently dominant strategies of gradual intervention. In such a case, lockdowns should be immediate and strict enough to stop the spread of the virus. It is also important to minimise their duration in order to decrease the negative economic consequences of the decreased activity. During the periods of mild restrictions, increases in the level of professional and private activity should be introduced gradually in order to decrease the rate of infections and to increase the time between consecutive lockdowns.

Figure 19 illustrates the scenario of introducing recurrent lockdowns in an economy. In the case of the first lockdown, we made the same assumptions as in the second scenario, which is presented in Section 4.2. Both lockdowns were introduced as a mobility restrictions that modified the grid and the interactions within a neighbourhood. The grid was also dynamically optimised throughout the simulation run. We assumed that the lockdown effect would be perpetuated by a part of a society, so their mobility was lower for some time despite the opening up of branches of the economy. During this period, the number of cases and mortality were low, and productivity was higher. After the transition period, when the mobility of agents increased, the number of infected also increased, which in turn forced the introduction of another lockdown. In this scenario the productivity of a healthy agent was not constant and was lower than one during the lockdown and the open-up phases. As in case of the second scenario, the productivity differential also reflected the varying degrees of the impact of the pandemic on the relevant sectors of the economy. We accept the possibility that this effect might not be exactly the same in the event of a subsequent lockdown (it may affect the shape of the productivity curve in the open-up phase). The open-up phase of the second lockdown was carefully planned and the shape of the curve reflected a strategy of closing and gradual opening sectors of the economy.

The macroeconomic consequences of recurrent lockdowns are depicted in Figure 20. The outcomes prove that consecutive lockdowns only resulted in temporary economic downturns of a limited duration. Monthly periods of a strict decrease in economic activity combined with a gradual open-up phase resulted in an approximately 4.5-month decrease in economic activity below its steady state level. What is important, is that after the lockdown phase, there was a period of increased economic activity. This result might play crucial role in assessing the proposed strategy as it permits an economy to make up for some of the losses during an epidemic episode. Such a turn of events might play an important role in ensuring the accumulation of reserves, which will help companies to survive further lockdowns. This feature of the recursive lockdown strategy distinguishes it from the scenarios that assumed a lack of targeted intervention, that were presented in Section 4, which would result in a permanent decrease of economic activity that lasted throughout the entire analysed period. As such, when rationally used and properly structured, a lockdown strategy might be more convenient for companies than initially thought.

The chances of success under a recursive lockdown strategy might be boosted significantly if the government introduced some additional provisions that were not yet included in the macroeconomic model presented above. Firstly, according to the rational expectations hypothesis when planning their economic activity, people use all of the available information. If so, open adoption and commitment to the proposed policy by the government might result in economic entities being better prepared for the lockdown phase. A public presentation of the draft lockdown schedules would allow entities to squeeze their actions within the mild restrictions phases in order to acquire reserves for the periods of decreased activity. Knowing that a lockdown is a temporary and strictly controlled situation will make decisions about the future of economic entities less uncertain, which would translate into a lower level of volatility in the macroeconomic categories and a lower cost of an epidemic.

Secondly, the model does not yet account for the role of fiscal policy, which could be an important source of economic stimulation in the lockdown periods. Wisely framed programmes of financial relief could decrease potential number of firm bankruptcies, while employment support programmes that bind employment subsidies with restrictions for dismissing employees could limit the volatility that is observed in the labour market. Such an approach could have a decisive impact on decreasing the social costs of a pandemic episode and could play an important role in maintaining social mobilisation in the fight against the disease.

Thirdly, the current version of the model ignores the costs of layoffs including the termination periods in labour contracts and severance payments. The same is true regarding the costs of hiring new employees during the periods of increased activity. In the absence of the aforementioned features, the model might overvalue the potential benefits of firing unproductive workers. As a result, the observed reactions of the employment and unemployment rates might overestimate the negative effects in labour market of lockdown episodes.

Finally, it should be noted that the model still lacks some of the features that might potentially increase the scale of the negative consequences of the lockdown policy. The most important of these is the lack of entry and exit of firms. In such a case, the depth of the recession that is caused by a lockdown might be slightly underestimated. The impact of that effect should, however, be counterbalanced by the contradictory tendencies that would result from the factors mentioned above as well as from the fact that according to the provided scenarios, we did only limit our analyses to a relatively short lockdown experience, which should be bearable for the majority of companies.

## 8. Conclusions

This paper presents the results of an examination of COVID-19 prevention and control schemes that was performed using the DSGE model with an agent-based epidemic component. The proposed methodology constitutes a new approach to the problem, and demonstrates its high potential for further use by providing a reasonable assessment of different epidemic scenarios. It shows its clear benefits compared to the traditional approach of epidemic models such as SIR model and its straightforward transformations as it permits introduction of much more elaborated dynamics of the disease, including the consequences of the spatial distribution of people and their social mobility. As a result, the methodology that was used in our paper enabled us to recreate a number of realistic prevention and control schemes and to assess their potential impact on a number of major macroeconomic indicators.

The research was designed in an effort to broaden the existing scientific perspective concerning the use and efficiency of epidemic prevention and control schemes. It addressed two of the most interesting economic questions that have been raised by the COVID-19 pandemic. The first concerned the efficacy of the use of lockdowns as an epidemic countermeasure, while the second tackled the issue of the efficient scale and composition of such a lockdown. The outcomes proved to be meaningful in both respects. Firstly, we have shown that the introduction of prevention and control schemes significantly decreases both the death toll and the overall level of economic disturbance, compared to the scenarios in which the persistent spread of COVID-19 is permitted. The decrease in economic activity in the case of lockdowns are deeper but more compact than in the case of the unlimited spread of the virus, in which the pace of economic growth and capital accumulation is permanently decreased, while societies have to cope with persistent high unemployment. Secondly, adopted methodology enabled us to compare the efficiency of the two major lockdown strategies that are currently being used: the one in which a lockdown is immediate and deep enough to limit the transmission of infections versus an approach in which a lockdown is introduced gradually. It turns out that the probability that gradual changes are deep enough to stop the spread of the coronavirus is relatively low, which results in extending the period that precedes an actual lockdown when an economy is already suppressed but when there is no improvements in terms of the pace of a virus spread. According to our results, this period is forlorn from an economic point of view and thus an economy would be better off if the lockdown were introduced in a decisive yet efficient manner. This observation is of major importance as it is contrary to the widespread belief that we should strive to keep an economy at least partially open as long as possible.

The outcomes of our research provide us with an interesting yet currently much overlooked conclusion concerning the advisable shape of an anti-COVID-19 policy. It turns out that lockdowns should not be perceived as a choice of last resort, but rather as a standard safety procedure that should be introduced when the number of infections exceeds a reasonable limit. Under certain conditions, they are not as damaging for an economy as it was earlier thought. Provided that people behave in a responsible way when coming out of a lockdown and maintain some standard safety provisions when they return to their professional activities, lockdowns permit us to significantly limit the duration of a period when the negative economic consequences of a spike in infections are experienced. If this is the case, we have reasons to presume that contingent on a proper informational strategy, a series of efficient lockdowns interspersed with periods of relatively normal activity could result in lower economic and social costs of pandemics than permitting it to spread freely across a society. This is mostly due to the fact that in such a case, we limit negative medium- and long-term consequences of an epidemic.

It should be noted that the results presented in this paper are still non-exhaustive and thus prone to some minor deficiencies as this publication only presents the introductory outcomes of the analyses that we believe were interesting enough to develop a more comprehensive research project to investigate the macroeconomic consequences of the COVID-19 pandemic. The model does not fully account for the complexity of the processes that are observed in an actual economy and society. In order to make our analyses more approachable we have decided not to include issues such as: the possible seasonality of infections, which might be an important factor that explains the dynamics of the pandemic that have been observed in the northern hemisphere; the problem of herd immunity, which might be an important yet, in our view, not yet fully scientifically confirmed aspect of COVID-19 containment policies (there is still insufficient scientific evidence on the persistence of the IgG and IgM antibodies after a successful COVID-19 recovery); the problem of the endogeneity of decisions concerning the labour market participation in the pandemic period that was raised by Eichenbaum et al. [1]; the dynamics of the labour market response, which occurs immediately after the shock, and does not account for the costs of hiring/firing workers, the termination periods in employment contracts and severance payments; a lack of the entry and exit effects of firms, which might affect the estimates concerning the depth of the economic downturn; the fiscal interventions that might possibly reduce the negative toll of the COVID-19 pandemic. Each of these issues constitutes a separate research topic that could result in a standalone research paper. Therefore, our results should be approached with due restraint.

## Figures and Tables

**Figure 1 entropy-22-01345-f001:**
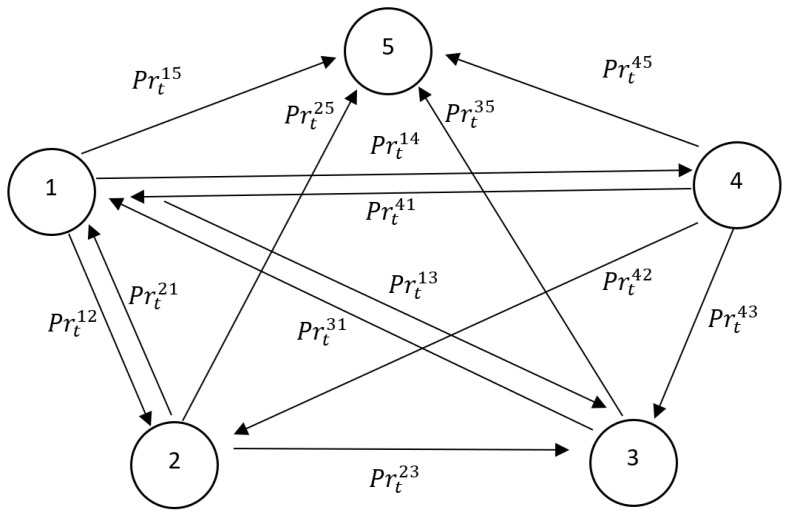
State transition probabilities in the agent-based epidemic component. *Health statuses: 1–healthy, 2–infected, 3–treated, 4–healthy individual in preventive quarantine, 5–deceased. $P_{ij}$–transition probability between states i and j, see Table 2 and Table 3.*

**Figure 2 entropy-22-01345-f002:**
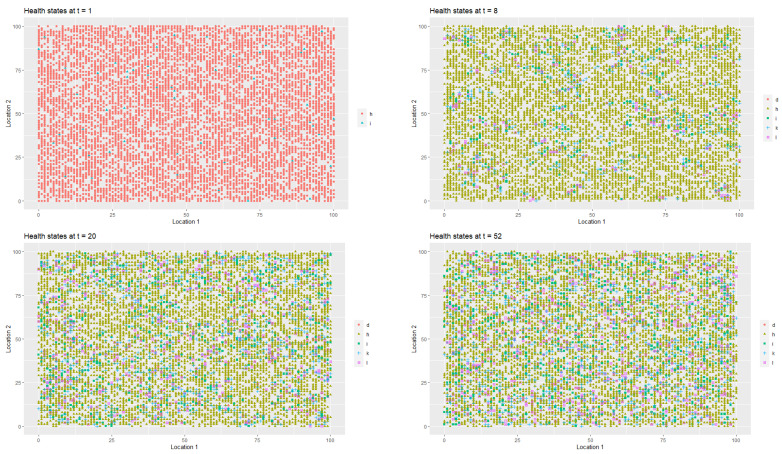
Scenario 1: Spatial-temporal spread of the coronavirus in a society. *States: Healthy (h), Infected (i), Treated (l), Preventive quarantine (k), Deceased (d).*

**Figure 3 entropy-22-01345-f003:**
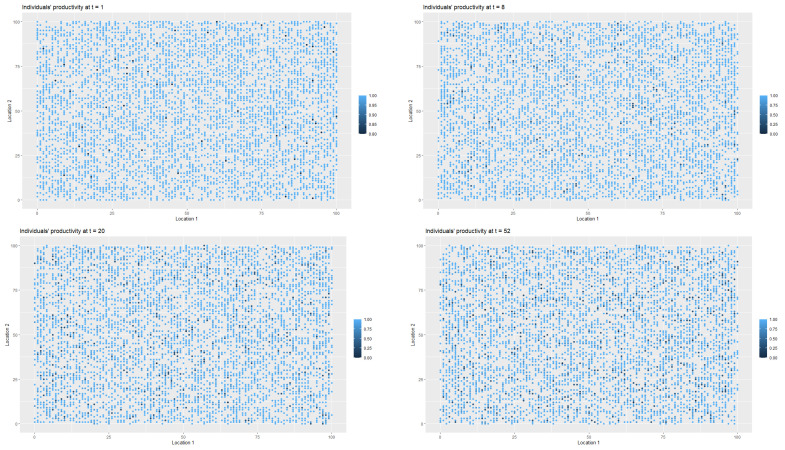
Scenario 1: Changes in agents’ productivity over time during the COVID-19 pandemic (people of working age only).

**Figure 4 entropy-22-01345-f004:**
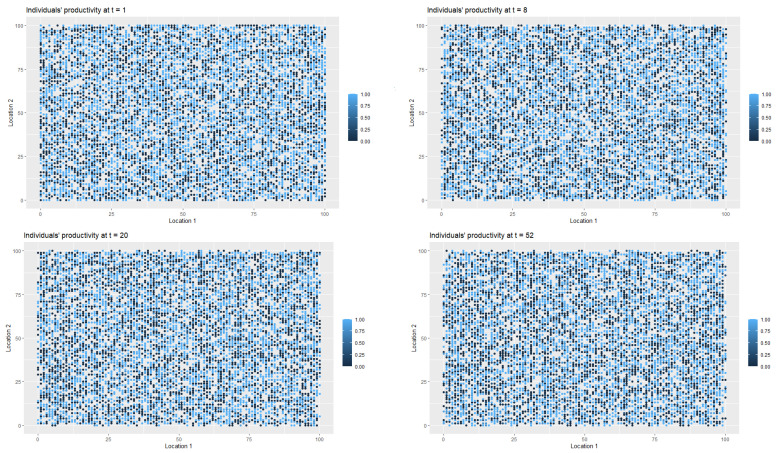
Scenario 1: Changes in agents’ productivity over time during the COVID-19 pandemic (including people in the pre-productive age and those who were retired and had by definition zero productivity).

**Figure 5 entropy-22-01345-f005:**
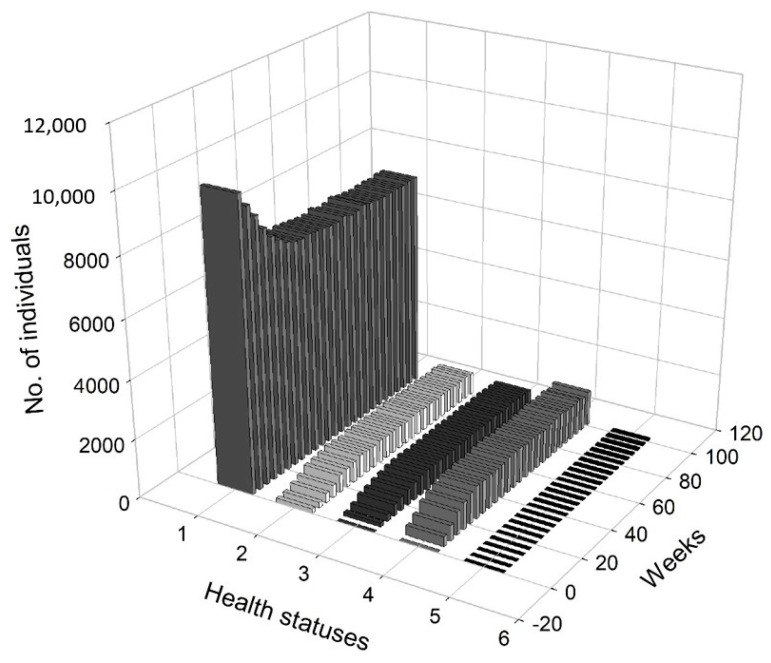
Scenario 1: 3D histogram of health statuses.

**Figure 6 entropy-22-01345-f006:**
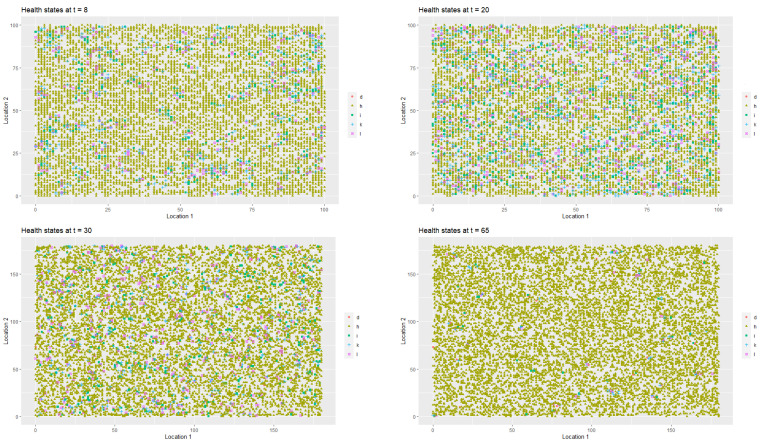
Scenario 2: Spatial-temporal spread of the coronavirus in a society (for first sub-scenario*). *States: Healthy (h), Infected (i), Treated (l), Preventive quarantine (k), Deceased (d)*; * See robustness checks in Section 6 for a further explanation.

**Figure 7 entropy-22-01345-f007:**
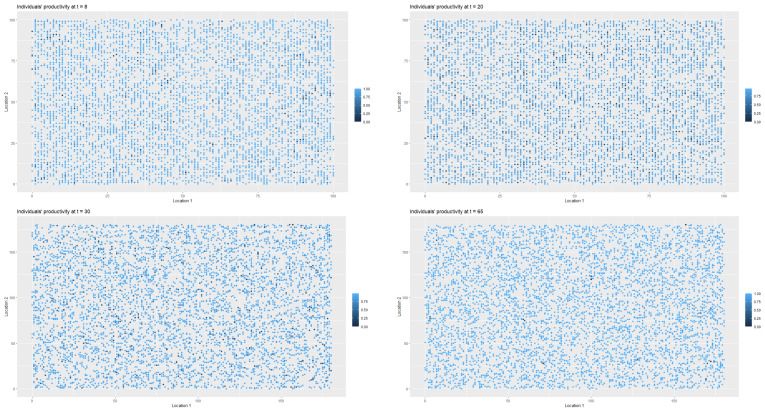
Scenario 2: Changes in individuals’ productivity over time during the COVID-19 pandemic for first sub-scenario (people of working age only). For further explanation see pp. 8–9.

**Figure 8 entropy-22-01345-f008:**
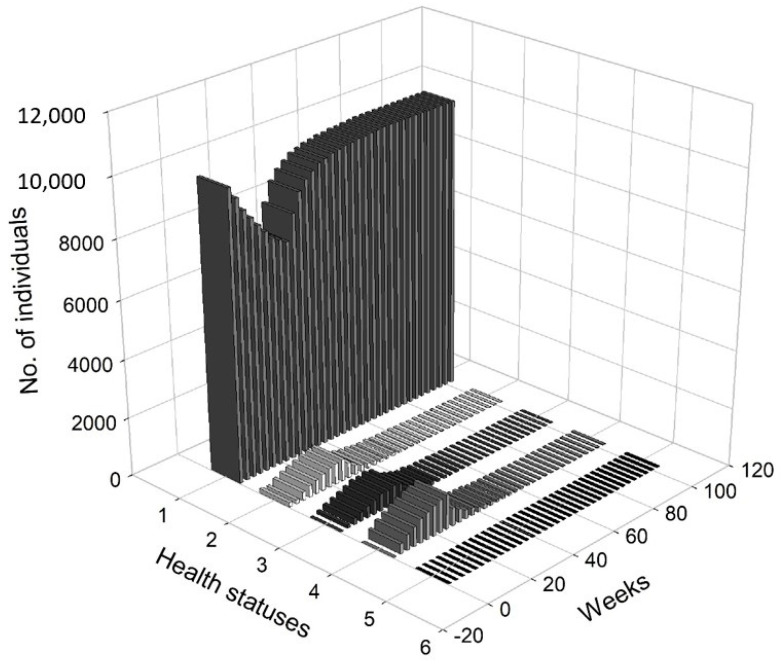
Scenario 2: 3D histogram of health statuses in the first sub-scenario.

**Figure 9 entropy-22-01345-f009:**
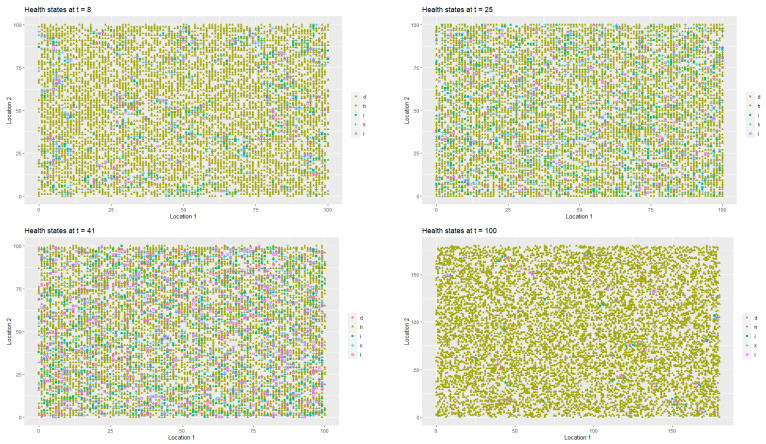
Scenario 3: Spatial-temporal spread of the coronavirus in a society. *States: Healthy (h), Infected (i), Treated (l), Preventive quarantine (k), Deceased (d).*

**Figure 10 entropy-22-01345-f010:**
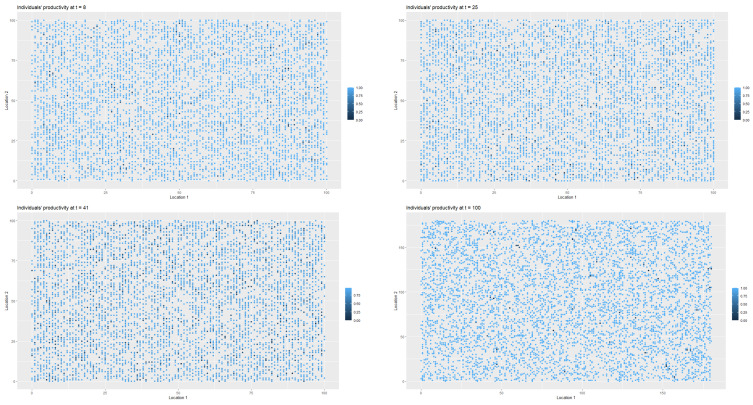
Scenario 3: Changes in individuals’ productivity over time during the COVID-19 pandemic (people of working age only). For further explanation see pp. 8–9.

**Figure 11 entropy-22-01345-f011:**
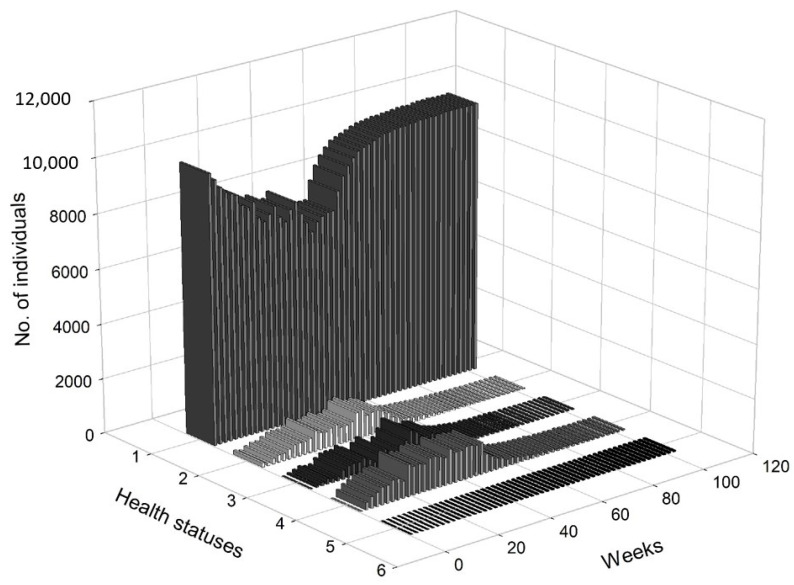
Scenario 3: 3D histogram of health statuses.

**Figure 12 entropy-22-01345-f012:**
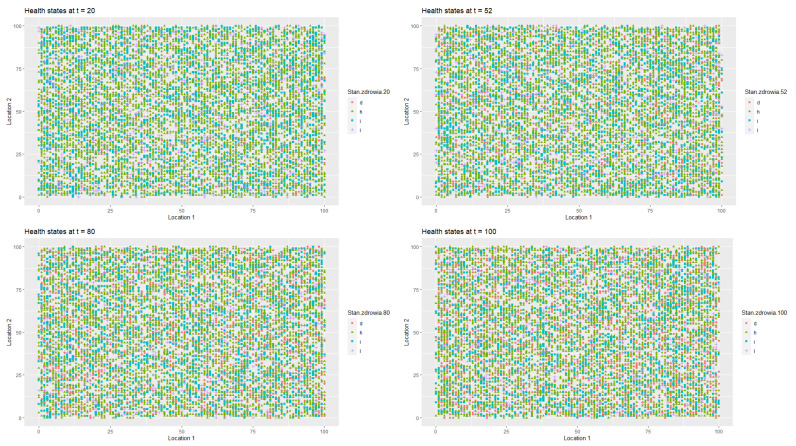
Scenario 4: Spatial-temporal spread of the coronavirus in a society. *States: Healthy (h), Infected (i), Treated (l), Preventive quarantine (k), Deceased (d)*.

**Figure 13 entropy-22-01345-f013:**
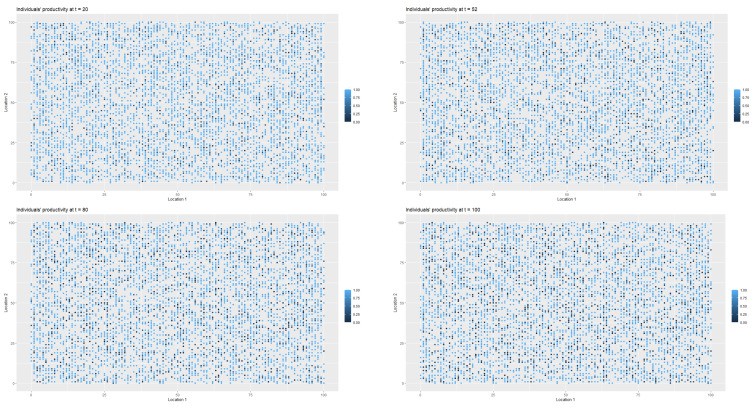
Scenario 4: Changes in individuals’ productivity over time during the COVID-19 pandemic (people of working age only). For further explanation see pp. 8–9.

**Figure 14 entropy-22-01345-f014:**
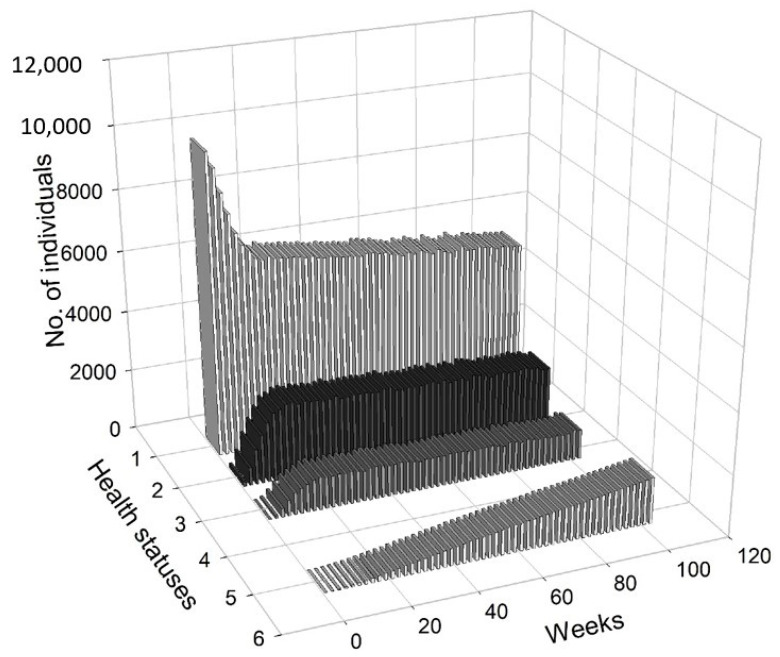
Scenario 4: 3D histogram of health statuses.

**Figure 15 entropy-22-01345-f015:**
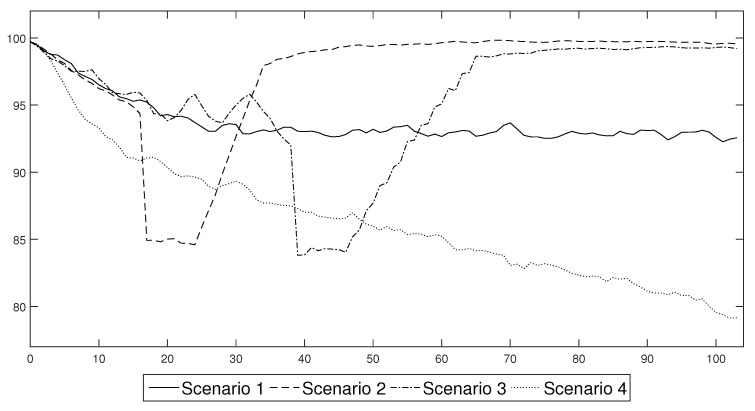
Aggregate labour productivity under different COVID-19 prevention and control schemes.

**Figure 16 entropy-22-01345-f016:**
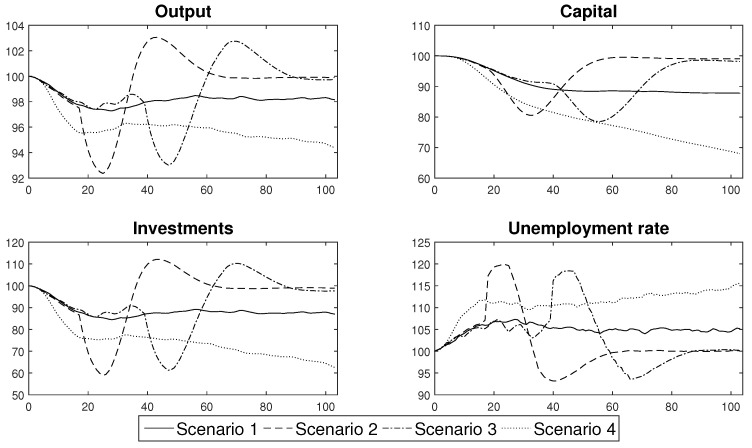
Conditional forecasts of the major macroeconomic indicators under different COVID-19 prevention and control schemes.

**Figure 17 entropy-22-01345-f017:**
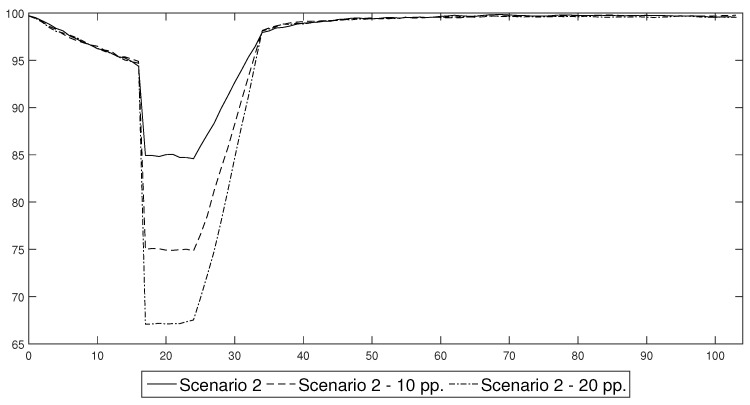
Labour productivity under Scenario 2–robustness tests.

**Figure 18 entropy-22-01345-f018:**
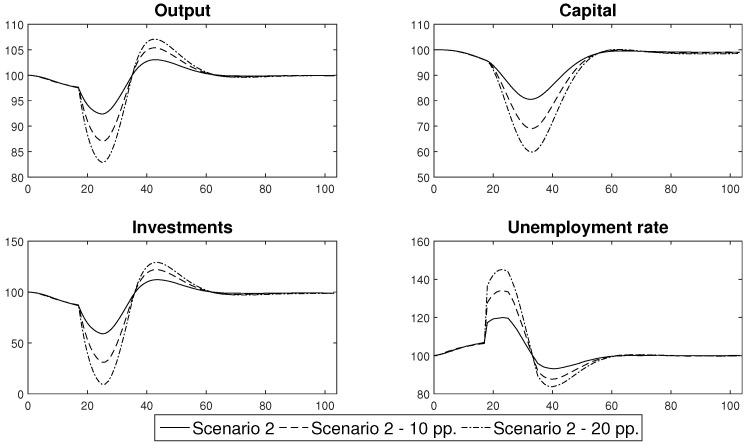
Conditional forecasts of the major macroeconomic indicators under Scenario 2–robustness tests.

**Figure 19 entropy-22-01345-f019:**
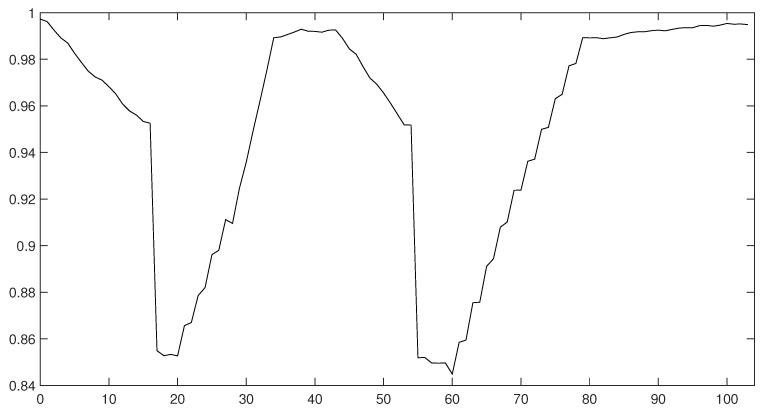
Labour productivity under recurrent lockdowns.

**Figure 20 entropy-22-01345-f020:**
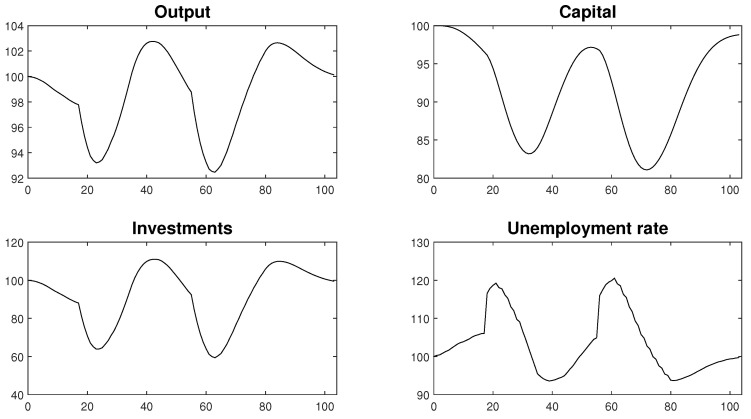
Conditional forecasts of the major macroeconomic indicators under recurrent lockdowns.

**Table 1 entropy-22-01345-t001:** Initial conditions/Parameters to be set.

Initial Conditions	Explanation	Restr.
*T*	Number of iterations (weeks).	≥0
$s_{t}^{Ind}$	Health status of the individual at time t=0 (1–healthy, 2–infected, 3–treated, 4–healthy individual in preventive quarantine, 5–deceased)	Int ∈{1,2,3,4,5}
${\left(Age\right)}_{t}^{Ind}$	Age of an individual at time t=0
$N^{Ind}$	Number of individuals at time t=0	Int ≥0
$K^{Ind}$	Number of infected individuals at time t=0 (including asymptomatically infected)	Int ≥0
$S_{t}\times S_{t}$	Dimensions of the grid at time *t**	Int ≥0
${\left(Ag\right)}_{t}^{1}$	Share of citizens of pre-working age at time *t*	∈〈0,1〉
${\left(Ag\right)}_{t}^{2}$	Share of citizens of working age at time *t*	∈〈0,1〉
${\left(Ag\right)}_{t}^{3}$	Share of retired individuals at time *t*	∈〈0,1〉
${\left(Wp\right)}_{t}^{Ind}$	Productivity of an individual at time t=0 (if healthy)	=1
${\left(Wp\right)}_{t}^{av\_inf}$	The productivity of an individual when infected at time *t* (the decline in productivity was estimated based on empirical data)	∈〈0,1〉
${\left(Wp\right)}_{t}^{av\_q}$	The productivity of an individual who is healthy and in quarantine at time *t* (the decline in productivity was estimated based on empirical data)	∈〈0,1〉
${\left(Wp\right)}_{t}^{av\_t}$	The productivity of an individual when treated or who is infected and in quarantine at time *t* (the decline in productivity was estimated based on empirical data)	∈〈0,1〉

* The dimensions are not constant in all scenarios for all *t*. In baseline scenario $S_{t}=S$.

**Table 2 entropy-22-01345-t002:** Probabilities set as parameters *.

Parameter	Explanation	Restr.
${\left(Pr\right)}_{t}^{12}$	The probability that a healthy individual (1) will become infected (2) at time *t*	∈(0,1)
${\left(Pr\right)}_{t}^{14}$	The probability that a healthy individual (1) will be in quarantine (although she is healthy) (4) at time *t*	∈(0,1)
${\left(Pr\right)}_{t}^{15}$	The probability that a healthy individual (1) will become infected and will die almost instantly (within week) (5)	∈(0,1)
${\left(Pr\right)}_{t}^{21}$	The probability that an infected individual (2) will become healthy (1)	∈(0,1)
${\left(Pr\right)}_{t}^{23}$	The probability that an infected individual (2) will be treated in a hospital or will stay in quarantine (3)	∈(0,1)
${\left(Pr\right)}_{t}^{25}$	The probability that an infected individual (2) dies (5)	∈(0,1)
${\left(Pr\right)}_{t}^{31}$	The probability that an infected individual in a hospital or quarantine (3) gets better (1)	∈(0,1)
${\left(Pr\right)}_{t}^{35}$	The probability that an infected individual in a hospital or quarantine (3) dies (5)	∈(0,1)
${\left(Pr\right)}_{t}^{41}$	The probability that a healthy individual in quarantine (4) will end the quarantine, that is, is healthy (1)	∈(0,1)
${\left(Pr\right)}_{t}^{43}$	The probability that a healthy individual in quarantine (4) will become infected during the quarantine and she is still in quarantine (but now is already infected) (3) at time *t*	∈(0,1)
${\left(Pr\right)}_{t}^{45}$	The probability that a healthy individual in quarantine (4) dies (5)	∈(0,1)

* Estimated on empirical data.

**Table 3 entropy-22-01345-t003:** Variables & Parameters that were computed by the program after each iteration.

Variable	Explanation	Restr.
${\left(Pr\right)}_{t}^{13}$	The probability that a healthy individual (1) will become treated in the hospital (or isolation) after becoming infected (3) at time *t*	∈(0,1)
${\left(Pr\right)}_{t}^{42}$	The probability that a healthy individual in quarantine (4) will become infected at the end of her quarantine (2)	∈(0,1)
*p*	Temporal variable (threshold probability 1)	∈(0,1)
*q*	Temporal variable (threshold probability 2)	∈(0,1)
*r*	Temporal variable (threshold probability 3)	∈(0,1)
$s_{t}^{Ind}$	Health status of the individual at time t>0 (1–healthy, 2–infected, 3–treated, 4–healthy individual in preventive quarantine, 5–deceased)	Int ∈{1,2,3,4,5}
${\left(Age\right)}_{t}^{Ind}$	Age of an individual at time t>0	≥0
${\left(Wp\right)}_{t}^{Ind}$	Productivity of an individual at time t>0	∈〈0,1〉

**Table 4 entropy-22-01345-t004:** Comparison of the calibration of scenarios 1–4.

Notation	Scenario 1	Scenario 2	Scenario 3	Scenario 4
*T*	104	104	104	104
$N^{Ind}$	10,000	10,000	10,000	10,000
$K^{Ind}$	150	150	150	150
$S_{t}\times S_{t}$	100×100 for all *t*	Dynamic adjustment	Dynamic adjustment	100×100 for all *t*
${\left(Ag\right)}_{t}^{1}$	0.181	0.181	0.181	0.181
${\left(Ag\right)}_{t}^{2}$	0.219	0.219	0.219	0.219
${\left(Ag\right)}_{t}^{3}$	0.6	0.6	0.6	0.6
${\left(Wp\right)}_{t}^{av\_h}$	1 for all *t*	Dynamic adjustment	Dynamic adjustment	1 for all *t*
${\left(Wp\right)}_{t}^{av\_inf}$	0.9	0.9	0.9	0.9
${\left(Wp\right)}_{t}^{av\_q}$	0.8	0.8	0.8	–
${\left(Wp\right)}_{t}^{av\_t}$	0.3	0.3	0.3	0.3
${\left(Pr\right)}_{t}^{12}$	0.03	0.03	Dynamic adjustment	0.2
${\left(Pr\right)}_{t}^{13}$	0.1	0.1	Dynamic adjustment	0
${\left(Pr\right)}_{t}^{15}$	0.00002	0.00002	Dynamic adjustment	0.00002
${\left(Pr\right)}_{t}^{21}$	0.6998	0.6998	Dynamic adjustment	0.6998
${\left(Pr\right)}_{t}^{24}$	0.2	0.2	Dynamic adjustment	0.2
${\left(Pr\right)}_{t}^{25}$	0.0002	0.0002	Dynamic adjustment	0.005
${\left(Pr\right)}_{t}^{41}$	0.6	0.6	Dynamic adjustment	–
${\left(Pr\right)}_{t}^{43}$	0.1	0.1	Dynamic adjustment	–
${\left(Pr\right)}_{t}^{45}$	0.0002	0.0002	Dynamic adjustment	–
${\left(Pr\right)}_{t}^{31}$	0.7	0.7	Dynamic adjustment	0.7
${\left(Pr\right)}_{t}^{35}$	0.0002	0.0002	Dynamic adjustment	0.002

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
