# Peer review of "To Freeze or Not to Freeze? Epidemic Prevention and Control in the DSGE Model Using an Agent-Based Epidemic Component"

_entropy, 2020, doi:10.3390/e22121345_

Round 1
Reviewer 1 Report
This is a very interesting and well-justified paper. I also agree with the conclusion and inferences made by the authors (especially line 694-697).
I do believe that it will earn many citations.
I also concur with authors' discussion, and I think that the conclusion regarding covid-19 and the associated interventions would be to follow a convex under uncertainty strategy. So, what leaders can do is minimize the risk to society and expose themselves to the risk of finding the best solution in economic terms. This is the "trial-and-error" (convexity) based on the performance function of a policy (Richard P. Feynman, What Do You Care What Other People: Think Further Adventures of a Curious Character (1988). Also, P. Samuelson exposed the process of trial and error which leads to the equilibrium of markets).
1). The paper is very technical and I would suggest the authors to clarify some terms. For instance, in section 3 they mention Moore neighborhood algorithm. I am not sure that the average reader (not only of Entropy's papers) knows about cellular automata. Hence, I would suggest to give a short definition in a footnote along with a reference.
2). I would also suggest to explain Figures 2, 3 and similar.
Author Response
Regarding the first review, the reviewer emphasized that the introduction has all of the necessary attributes and clearly sets out the main idea; the research design was appropriate and the methods adequately described. The conclusions were supported by the results. We have improved the presentation of results.
The reviewer noted that “English language and style are fine/minor spell check required”. However, we would like to emphasize that we decided to send our article to a professional editor (Ms. M. Simmons).
The first reviewer stated that our article was very interesting and well-justified. The reviewer agreed with the conclusions and inferences made by the authors (especially line 694-697, after changes: lines 727-730).
The first reviewer believed that the article would earn many citations. (S)he also concurred with authors' discussion, and thought that the conclusion regarding COVID-19 and the associated interventions would be to follow a convex under uncertainty strategy.
We would like to thank the reviewer for this comment. It allowed us to refine the most important section of the article, i.e. “Policy implications”. We might have used different words, but we definitely wanted to emphasize the goal in front of leaders, which is to minimize the risk to society, while it may require exposing themselves to the risk of finding the best solution in economic terms.
The first reviewer emphasized that the paper was technical and suggested the authors to clarify some terms (e.g. Moore neighbourhood, cellular automata models).
We added the simplest definition of Moore neighbourhood in the paper. It is the cell itself and the cells at a Chebyshev distance of 1 or saying it more straightforward: in cellular automata models, the Moore neighbourhood is defined on a two-dimensional square lattice (grid) and is composed of a central cell and the eight cells that surround it. We added the references on cellular automata models and the Moore neighbourhood.
The first reviewer also suggested the authors to explain Figures 2, 3 and similar.
We added the explanation (see pp. 8-9, lines 256-270). In addition, we removed agents in the pre-productive age and those who were retired and had by definition zero productivity, from Figures: 3, 7, 10, 13. In order to allow the reader to compare, in Fig. 4, we described the movement of agents and productivity of the whole society (agents in the pre-productive age, people in working age and those who were retired).
Moreover, we adjusted figures: 5, 8, 11 and 14 for clarity. We also added the clarification below the figure 1 (in the footnote).
Please find enclosed the whole response for comments addressed by both reviewers.
Thank you.
Kind regards,
The authors.

Reviewer 2 Report
In this article the authors present their findings after suggesting and investigating a new model of the economic and health effects of various lockdown policies in the face of COVID-19 population infection. The novelty comes from the agent-based model (ABM) approach as an informative model for the popular dynamic stochastic general equilibrium (DSGE) model. The authors not only outline their findings, but conclude with the policy implications of their findings compared with current approaches. The work is timely and an interesting exploration. I especially appreciated the candid section at the end about caveats and reservations, as always exist with models.
Intellectual Comments
My cursory knowledge of economic modeling leaves me with 1 question from the very beginning: If the model you investigate explores changes in policy using DSGE, then how do you account for the common criticism that DSGE is incredibly sensitive (in the wrong ways) to changes in policy? Addressing the criticism of DSGE known as the "Lucas Critique (1976)" with a single sentence would suffice.
Visual Comments
Figure 1 provides very little information, even with the caption. Nearly everything except the directionality of the arrows has been abstracted away. If you add textual labels to the nodes, then the reader can at least infer Table 2 simply by looking at the figure, which would be a huge information gain for this figure.
Figure 2-12 2D grid plots. This is an onslaught of information for the reader, not all of it relevant, and I found myself trying to make statistical comparisons by eye, which is always a sign that perhaps a more statistically concise figure would be more appropriate. Consider as a thought exercise if you could only show a single line plot, what information are you trying to convey from the original Fig 2? and what statistical aggregation would you employ?
Writing Comments
The good: I could always figure out what the author probably intended.
The bad: The paper reads like it was written by 2 different people with very different levels of writing skill, in a disjointed fashion. The grammar needs a great amount of work to be concise, clear, and more easily digested.
We have similar asymmetric author situations that arise during the writing process at our university. The way we solve this is through hiring a professional editor -- this article is an excellent candidate for the work of a professional editor and would go a long way to improving the article up to the standards of an Entropy publication. Most universities or departments offer this service to their faculty or will reimburse such expenses. I heavily suggest you make this investment one way or the other: you will find the before-after comparison highly enlightening.
Author Response
The second reviewer emphasized that the authors presented their findings after suggesting and investigating a new model of the economic and health effects of various lockdown policies in the face of COVID-19 population infection. The novelty comes from the agent-based model (ABM) approach as an informative model for the popular dynamic stochastic general equilibrium (DSGE) model. The authors not only outlined their findings, but concluded with the policy implications of their findings compared with current approaches.
The reviewer perceived our work as timely and an interesting exploration. The second reviewer especially appreciated the candid section at the end about caveats and reservations, as always exist with models.
We would like to thank the second reviewer. We greatly appreciated his/her comments and suggestions.
Regarding “Intellectual comments”, we added the following sentences (see p. 25 lines 646-652):
“Due to the fact that this DSGE model is microfounded and have deep parameters that are invariant to changes in economic policy, in principle it is not subject to the Lucas critique. However, for more general discussion about DSGE models and the Lucas critique, see Hurtado (2014). The resistance to the Lucas and Velupillai's critiques of core and periphery models (in particular, DSGE and ABM models) was also studied in Kaszowska et al. (2019).”
We would like to thank the reviewer for the visual comments. We added the following sentences below Figure 1 (p.6):
“*Health statuses: 1 - healthy, 2 - infected, 3 - treated, 4 - healthy individual in preventive quarantine, 5 - deceased. P_ij - transition probability between states i and j, see Tables 2 & 3”.
Regarding the comment to Figure 2-12 2D grid plots, we would like to emphasize that the method of data presentation is analogous to the one used in case of segregation models such as the Schelling model. The grid presents the locations of agents and their health statuses [or labour productivity] for a given week (iteration).
However, we agree that we should have explained it better. We provided an explanation in the article and we emphasized that data depicted in Figures 3, 7, 10 and 13 was then used to obtain Figure 15 (Fig. 15. Aggregate labour productivity under different COVID-19 prevention and control schemes -> a single line plot represents a given scenario).
We added the explanation to Figures 2 & 3 (see pp. 8-9, lines 256-270). In addition, we removed agents in the pre-productive age and those who were retired and had by definition zero productivity, from Figures: 3, 7, 10, 13. To allow the reader to compare, in Fig. 4, we described the movement and productivity of the whole society (in the pre-productive age, in working age and those who were retired).
Moreover, we adjusted figures: 5, 8, 11 and 14 (we used other statistical toolbox).
Regarding “Writing Comments”, we do apologize. Please find attached a new version, which was proofread by M. Simmons, a professional editor.
Please find enclosed the whole response for comments addressed by both reviewers.
Thank you.
Kind regards,
The authors
